# Bounding generalization error with input compression: An empirical study with infinite-width networks

**Angus Galloway**[1,2]                                                    *gallowaa@uoguelph.ca*

**Anna Golubeva**[3,4]                                                     *golubeva@mit.edu*

**Mahmoud Salem**[1,2]                                                     *msalem04@uoguelph.ca*

**Mihai Nica**[5,2]                                                        *nicam@uoguelph.ca*

**Yani Ioannou**[6]                                                        *yani.ioannou@ucalgary.ca*

**Graham W. Taylor**[1,2,7]                                                *gwtaylor@uoguelph.ca*

*1. School of Engineering, University of Guelph, ON, Canada*
*2. Vector Institute for Artificial Intelligence, ON, Canada*
*3. The NSF AI Institute for Artificial Intelligence and Fundamental Interactions*
*4. Department of Physics, Massachusetts Institute of Technology, MA, USA*
*5. Mathematics & Statistics, University of Guelph, ON, Canada*
*6. Schulich School of Engineering, University of Calgary, AB, Canada*
*7. Canada CIFAR AI Chair*

**Reviewed on OpenReview:** *https://openreview.net/forum?id=jbZEUtULft*

## Abstract

Estimating the Generalization Error (GE) of Deep Neural Networks (DNNs) is an important task that often relies on availability of held-out data. The ability to better predict GE based on a single training set may yield overarching DNN design principles to reduce a reliance on trial-and-error, along with other performance assessment advantages. In search of a quantity relevant to GE, we investigate the Mutual Information (MI) between the input and final layer representations, using the infinite-width DNN limit to bound MI. An existing input compression-based GE bound is used to link MI and GE. To the best of our knowledge, this represents the first empirical study of this bound. In our attempt to empirically stress test the theoretical bound, we find that it is often tight for best-performing models. Furthermore, it detects randomization of training labels in many cases, reflects test-time perturbation robustness, and works well given only few training samples. These results are promising given that input compression is broadly applicable where MI can be estimated with confidence.

## 1 Introduction

Generalization Error (GE) is the central quantity for the performance assessment of Deep Neural Networks (DNNs), which we operationalize as the difference between the train-set accuracy and the test-set accuracy[1]. Bounding a DNN's GE based on a training set is a longstanding goal (Jiang et al., 2021) for various reasons: i) Labeled data is often scarce, making it at times impractical to set aside a representative test set. ii) The ability to predict generalization is expected to yield overarching design principles that may be used for Neural Architecture Search (NAS), reducing a reliance on trial-and-error. iii) Bounding the error rate is helpful for model comparison and essential for establishing performance guarantees for safety-critical applications. In

---

[1]GE is also referred to as *generalization gap*. Note that some use "generalization error" as a synonym for "test error".

contrast, the test accuracy is merely a single performance estimate based on an arbitrary and finite set of examples. Furthermore, the *adversarial examples* phenomenon has revealed the striking inability of DNNs to generalize in the presence of human-imperceptible perturbations (Szegedy et al., 2014; Biggio & Roli, 2018), highlighting the need for a more specific measure of *robust* generalization.

Various proxies for DNN complexity which are assumed to be relevant to GE—such as network depth, width, $\ell_p$-norm bounds (Neyshabur et al., 2015), or number of parameters—do not consistently predict generalization in practice (Zhang et al., 2021). In search of an effective measure to capture the GE across a range of tasks, we investigate the Mutual Information (MI) between the input and final layer representations, evaluated solely on the training set. In particular, we empirically study the Input Compression Bound (ICB) introduced by (Tishby, 2017; Shwartz-Ziv et al., 2019), linking MI and several GE metrics. An emphasis on *input* is an important distinction from many previously proposed GE bounds (e.g., Zhou et al. (2019)), which tend to be *model*-centric rather than *data*-centric.

We use *infinite ensembles of infinite-width networks* (Lee et al., 2019), as in deterministic DNNs the MI quantity we examine is ill-defined (Goldfeld et al., 2019). Infinite-width networks correspond to *kernel regression* and are simpler to analyze than finite-width DNNs, yet they exhibit double-descent and overfitting phenomena observed in Deep Learning (DL) (Belkin et al., 2019). For these reasons, Belkin et al. (2018) suggested that understanding kernel learning should be the first step taken towards understanding generalization in DL. To this end, we evaluate the ICB with respect to three criteria:

(1) First, we asess the extent to which the bound holds in practice. To do so, we evaluate the GE of a variety of models, composed by many different metaparameters of the neural architecture and training procedure. We then compare the GE to the theoretical GE bound given by ICB. We show that ICB contains the GE at the expected frequency for four of five datasets. We identify several factors that influence ICB success, including a training-label randomization test inspired by Zhang et al. (2017).

(2) Next, we analyze whether the ICB is sufficiently small for useful model comparisons. If a theoretical GE bound exceeds 100% in practice, it is said to be *vacuous*. As we study binary classification tasks, we additionally require that the bound be less than 50% for models with non-trivial GE. We show that ICB is often sufficiently close to the empirical GE, and thus presents a *non-vacuous* bound. These results are obtained with 2000 or fewer training samples.

(3) Last, we assess the ability of ICB to predict changes in GE under different metaparameter interventions. The ICB accurately predicts changes in GE as the training time, number of training samples, and explicit parameter regularization vary; ICB was less consistent when the activation function, input data, and depth vary, depending on the dataset.

Beyond these three main desiderata for generalization bounds, we show advantages in reducing ICB even when the GE is small. Reducing ICB on *natural* training labels prevents models from fitting *random* labels, and conversely, ICB *increases* when models are trained on *random* versus *natural* training labels (Zhang et al., 2017; 2021). Finally, we show that ICB is predictive of test-time perturbation robustness (Goodfellow et al., 2015; Gilmer et al., 2019), without assuming access to a differentiable model.

## 2 Background

We make use of an information-theoretically motivated generalization bound, the ICB, to establish an overlooked link between MI and GE. The bound seems to have first appeared in a lecture series (see, e.g., Tishby (2017)), later in a pre-print (Shwartz-Ziv et al., 2019)[Thm. 1] and more recently in a thesis (Shwartz-Ziv, 2021)[Ch. 3]. To the best of our knowledge the bound has not yet been studied empirically.

### 2.1 Mutual information in infinite-width networks

The MI between two random variables $X$ and $Z$ is defined as

$$I(X; Z) \equiv \sum_{x,z} p(x,z) \log \frac{p(x,z)}{p(x)p(z)} = \mathbb{E}_{p(x,z)} \left[ \log \frac{p(z|x)}{p(z)} \right], \tag{1}$$

where we used Bayes' rule to obtain the expression on the right and introduced $\mathbb{E}_{p(x,z)}[\cdot]$ to denote the average over $p(x,z)$. In our case, $X$ denotes the input, and $Z$ the input representation which is taken as the Neural Network (NN) output. Since the marginal $p(z)$ is unknown, we use an unnormalized multi-sample "noise contrastive estimation" (InfoNCE) variational bound. The InfoNCE procedure was originally proposed for unsupervised representation learning (van den Oord et al., 2018), which also serves as a lower bound on MI (Poole et al., 2019). In van den Oord et al. (2018), the density ratio $p(z|x)/p(z)$ was learned by a NN. Instead, following Shwartz-Ziv & Alemi (2020), we use infinite ensembles of infinitely-wide NNs, which have a conditional Gaussian predictive distribution: $p(z|x) \sim \mathcal{N}(\mu(x,\tau), \Sigma(x,\tau))$ with $\mu, \Sigma$ given by the Neural Tangent Kernel (NTK) and Neural Network Gaussian Process (NNGP) kernel (Jacot et al., 2018). The predictive distribution also remains Gaussian following $\tau$ steps of Gradient Descent (GD) on the Mean-Squared Error (MSE) loss. The conditional Gaussian structure given by NTK may be supplied in the InfoNCE procedure, yielding MI bounds free from variational parameters. Specifically, we use the "leave one out" upper bound (Poole et al., 2019) on MI to conservatively bound MI:

$$I(X;Z) \leq \mathbb{E}\left[\frac{1}{N}\sum_{i=1}^{N}\log\frac{p(z_i|x_i)}{\frac{1}{N-1}\sum_{j\neq i}p(z_i|x_j)}\right] = I_{\text{UB}}. \tag{2}$$

A lower bound on MI, $I_{\text{LB}}$, of a similar form as Eq. (2) is also available (Eq. 6, Appendix A.2). We verified that both bounds yield similar results for the training regime in which we apply them (Fig. A5). See van den Oord et al. (2018) and Poole et al. (2019) for formal derivations of Eq. (2) and (6). These MI bounds must be computed on the training set only to evaluate a *generalization* bound.

## 2.2 Input compression bound

Here, we provide an intuitive explanation of the ICB building on existing results and using information theory fundamentals (Cover & Thomas, 1991). A more formal derivation including a proof can be found in Shwartz-Ziv et al. (2019)[Appendix A]. We begin with the conventional Probably Approximately Correct (PAC) GE bound, which plays a central role in early mathematical descriptions of machine learning (Shalev-Shwartz & Ben-David, 2014). It is assumed that a model receives a sequence of examples $x$, each labeled with the value $f(x)$ of a particular target function, and has to select a hypothesis that approximates $f$ well from a certain class of possible functions. By relating the hypothesis-class cardinality $|\mathcal{H}|$ and the number of training examples $N_{\text{trn}}$, one obtains the following bound on the GE:

$$\text{GE} < \sqrt{\frac{\log(|\mathcal{H}|) + \log(1/\delta)}{2N_{\text{trn}}}} \tag{3}$$

where the confidence parameter $\delta \in (0,1)$ indicates the probability with which the bound holds w.r.t. the random training sample. The key term in this bound is the hypothesis-class cardinality, the *expressive power* of the chosen ansatz. For a finite $\mathcal{H}$, it is simply the number of possible functions in this class; when $\mathcal{H}$ is infinite, a discretization procedure is applied in order to obtain a finite set of functions. For NNs, $|\mathcal{H}|$ is usually assumed to increase with the number of trainable parameters. The bound (3) states that generalization is only possible when the expressivity is outweighed by the size of the training set, in line with the well-known bias-variance trade-off of statistical learning theory. Empirical evidence, however, demonstrates that this trade-off is qualitatively different in deep learning, where generalization tends to improve as the NN size increases even when the size of the training set is held constant. The key idea behind the ICB is to shift the focus from the hypothesis to the *input space*. For instance, consider binary classification where each of the $|\mathcal{X}|$ inputs belongs to one of two classes. The approach that leads to bound (3) reasons that there are $2^{|\mathcal{X}|}$ possible label assignments, only one of which is true, and hence a hypothesis space with $2^{|\mathcal{X}|}$ Boolean functions is required to guarantee that the correct labeling can be learned. The implicit assumptions made here are that all inputs are fully distinct and that all possible label assignments are equiprobable. These assumptions do not hold true in general, since classification fundamentally *relies* on similarity between inputs. However, the notion of similarity is data-specific and a priori unknown; thus, the uniformity assumption is required when deriving a general statement.

The approach behind ICB circumvents these difficulties altogether by applying information theory to the process of NN learning. First, note that solving a classification task involves finding a suitable partition of the

input space by class membership. DNNs perform classification by creating a representation $Z$ for each input $X$ and progressively coarsening it towards the class label, which is commonly represented as an indicator vector. The coarsening procedure is an inherent property of the NN function, which is implicitly contained in $Z$. By construction, the NN implements a partitioning of the input space, which is adjusted in the course of training to reflect the true class membership. In this sense, the cardinality of the hypothesis space reduces to $|\mathcal{H}| \approx 2^{|\mathcal{T}|}$, where $|\mathcal{T}|$ is the number of class-homogeneous clusters that the NN distinguishes. To estimate $|\mathcal{T}|$, the notion of *typicality* is employed: *Typical* inputs have a Shannon entropy $H(X)$ that is roughly equal to the average entropy of the source distribution and consequently a probability close to $2^{-H(X)}$. Since the typical set has a probability of nearly 1, we can estimate the size of the input space to be approximately equal to the size of the typical set, namely $2^{H(X)}$. Similarly, the average size of each partition is given by $2^{H(X|Z)}$. An estimate for the number of clusters can then be obtained by $|\mathcal{T}| \approx 2^{H(X)}/2^{H(X|Z)} = 2^{I(X;Z)}$, yielding a hypothesis class cardinality $|\mathcal{H}| \approx 2^{2^{I(X;Z)}}$. With this, the final expression for the ICB is:

$$\text{GE}_{\text{ICB}} < \sqrt{\frac{2^{I(X;Z)} + \log(1/\delta)}{2N_{\text{trn}}}}, \tag{4}$$

where it is assumed that $X$ is a $d$-dimensional random variable that obeys an ergodic Markov Random Field (MRF) probability distribution, asymptotically in $d$ (common for signal and image data, see e.g., Murphy (2012)[Ch. 19]). Unfortunately, it is impossible to check this assumption directly because it involves the data-generating process, which we can not access based on finitely many samples (i.e., the training set). We therefore treat ICB as a tool, and empirically test how useful this tool is in practice. We comment on the ergodic MRF assumption in Appendix A.1. We only evaluate ICB when we can obtain a confident estimate of $I(X;Z)$. For this we require a tight sandwich bound on $I(X;Z)$ with $I_{\text{UB}} \approx I_{\text{LB}}$. We discard samples where $I_{\text{UB}}(X;Z) > \log(N_{\text{trn}})$, since $I_{\text{LB}}(X;Z)$ cannot exceed $\log(N_{\text{trn}})$. See Fig. A5 for typical $I_{\text{UB}}, I_{\text{LB}}$ values during training and samples to discard. Note that the units for $I(X;Z)$ in ICB are *bits*.

## 3  Experiments

Our experiments are structured around three main questions: 1) To what extent does ICB bound GE in practice (§4.1), including for random labels (§4.2)? 2) Is the ICB close enough to the empirical GE to be useful for model comparisons (§4.3)? 3) To what extent does ICB predict GE with respect to different metaparameter interventions (§4.4)?

We focus on binary classification like much of the generalization literature, which also enables us to more efficiently evaluate MI bounds by processing kernel matrices that scale by $N_{\text{trn}}^2$ rather than $(k \times N_{\text{trn}})^2$ for $k$ classes. Aside from this computational advantage, our methodology extends to the multi-class setting. We conduct experiments with five standard datasets: `MNIST` (LeCun & Cortes, 1998), `FashionMNIST` (Xiao et al., 2017), `SVHN` (Netzer et al., 2011), `CIFAR-10` (Krizhevsky, 2009), and `EuroSAT` (Helber et al., 2018; 2019). These datasets are intended to be representative of low to moderate complexity tasks and make it tractable to train thousands of models (Jiang* et al., 2019). Experiments with `EuroSAT` further demonstrate how the method scales to 64-by-64 pixel images. For each of the image datasets, we devise nine binary classification "tasks" corresponding to labels "$i$ versus $i + 1$" for $i \in \{0, \dots, 8\}$. This sequential class ordering ensures each label is used once, and is otherwise an arbitrary choice.

We use metaparameters that are common to deep learning, with the exception of "diagonal regularization", which is specific to the NTK, denoted by $\mathcal{K}$. The regularized NTK is defined as: $\mathcal{K}_{\text{reg}} = \mathcal{K} + \lambda \frac{\text{Tr}(\mathcal{K})}{N_{\text{trn}}} I$, where $\lambda$ is a coefficient that controls the amount of regularization. This is analogous to $\ell_2$ regularization of finite-width DNNs, only we penalize the parameters' distance w.r.t. their initial values instead of w.r.t. the origin (Lee et al., 2020). We train a population of models by selecting metaparameters as follows: the total number of layers $D \in \{1, 2, 3, 4, 5\}$, diagonal regularization coefficients $\lambda \in \{10^0, 10^{-1}, 10^{-2}, 10^{-3}, 10^{-4}\}$, and activation functions $\phi(\cdot) \in \{\texttt{ReLU}(\cdot), \texttt{Erf}(\cdot)\}$. For fully-connected Multi-Layer Perceptron (MLP) models we select the number of training samples, $N_{\text{trn}} \in \{1000, 1250, 1500, 1750, 2000\}$, whereas for Convolutional Neural Networks (CNNs) we use $N_{\text{trn}} \in \{1000, 1500\}$ for computational reasons. Test sets have constant $N_{\text{tst}} = 2000$. We do not vary the learning rate or mini-batch size, as the infinite-width networks are trained by full-batch GD, for which the training dynamics do not depend on the learning rate once below a critical

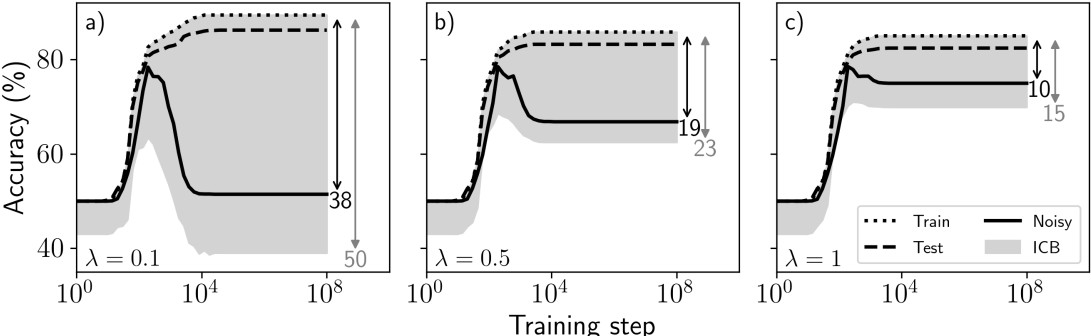

Figure 1: **The ICB may reflect robust GE when it is loose w.r.t. standard GE.** The ICB is plotted as a grey shaded band underneath the training accuracy indicating the range of test accuracy compatible with the theoretical bound. Performance metrics are evaluated for a `EuroSAT Pasture` versus `Sea-Lake` binary classification task using 500 training samples and 2000 test samples and different regularization levels in (a)–(c). At low regularization $\lambda = 0.1$ (a), the ICB is vacuous with respect to standard generalization beyond $10^4$ training steps, but reflects the poor robust generalization for the "Noisy" test set with AWGN. Increasing the regularization to $\lambda = 0.5$ (b) and $\lambda = 1.0$ (c) reduces ICB and the AWGN GE. Arrows indicate the steady-state AWGN GE (black), and ICB (grey) along with their respective values. See Fig. A5 for the corresponding upper and lower $I(X; Z)$ bounds for this experiment.

stable value (Lee et al., 2019). A nominal learning rate of 1.0 was used in all cases and found to be sufficient.[2] Models were evaluated at five different training times, $t \in \{10^2, 10^3, 10^4, 10^5, 10^6\}$, which contained most of the variation in GE. Training for less than $t = 10^2$ steps typically resulted in a small GE, as both training and test accuracy were near random chance or increasing in lockstep. In terms of steady-state behaviour, GE was often stable beyond $t = 10^6$. Furthermore, $t = 10^6$ was found to be a critical time beyond which $I_{\mathrm{UB}}(X; Z)$ sometimes exceeded its upper confidence limit of $\log(N_{\mathrm{trn}})$, particularly for small $\lambda$ values where memorization (lack of compression) is possible. In total, there are 1250 unique metaparameter combinations for MLPs: $5\ (D) \times 5\ (\lambda) \times 2\ (\phi) \times 5\ (N_{\mathrm{trn}}) \times 5\ (t) = 1250$, and 500 for CNNs.

We measure correlations between ICB and standard GE, as well as *robust* GE. The latter is defined as the accuracy difference on the standard train set and the test set with Additive White Gaussian Noise (AWGN) (Gilmer et al., 2019). It can be shown that a classifier's error rate for a test set corrupted by AWGN determines the mean distance to the decision boundary for linear models (Fawzi et al., 2016) and serves as a useful guide for DNNs (Gilmer et al., 2019). For the AWGN we use a Gaussian variance $\sigma^2 = 1/16$ for `EuroSAT` and $\sigma^2 = 1/4$ for the other datasets.

## 4  Results

**Illustrative example** Before presenting the main results, we examine ICB for a `EuroSAT` classification task using only 500 training samples (Fig. 1). This is a challenging task, as tight MI and GE bounds are difficult to obtain for high-dimensional DNNs, particularly with few samples. For example, in (Dziugaite & Roy, 2017) 55000 samples were used to obtain a $\approx 20\%$ GE bound for finite-width DNNs evaluated on `MNIST`.

We evaluate ICB throughout training from the first training step ($t = 10^0$) until steady state when all accuracies stabilize ($t = 10^8$). Shortly after model initialization ($t = 10^0$ to $t = 10^1$) the ICB is $< 7\%$ (indicated by the height of the shaded region in Fig. 1) and the training and test accuracy are both at 50% (GE$= 0$). Here, ICB is non-vacuous, but also not necessarily interesting for this random-guessing phase. ICB increases as training is prolonged.[3] At low regularization (Fig. 1 a), the ICB ultimately becomes vacuous (ICB $\approx 50\%$) around $10^4$ steps. However, although ICB is vacuous with respect to *standard* generalization in

---

[2]This was the default setting for the `neural_tangents` software library (Novak et al., 2020).

[3]It may not be obvious that ICB increases monotonically with training steps as the training accuracy also increases.

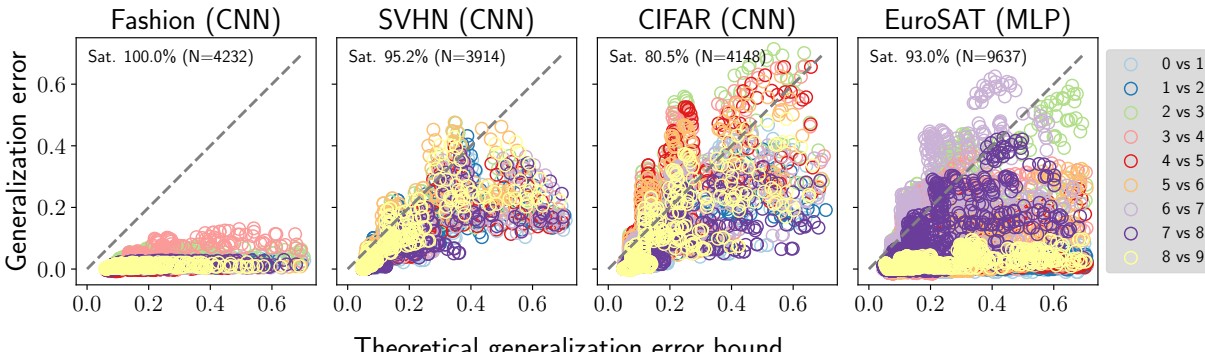

Figure 2: **The ICB often bounds GE for most datasets.** The ICB ("Theoretical generalization error bound") is plotted versus GE (MSE) for four datasets. The ICB satisfaction rate is annotated in the top left corner of each plot with format "Sat. % ($N$)", where $N$ denotes the number of valid experiments. Each binary classification task is assigned a unique colour. See Fig. A6 for equivalent plot using the 0–1 loss.

Table 1: Overall ICB satisfaction rate for two different loss functions, with the number of valid experiments ($N$) in brackets. The `MNIST` and `FashionMNIST` datasets were both at 100% and omitted for brevity.

| Loss | `SVHN` (CNN) | `CIFAR` (CNN) | `EuroSAT` (MLP) |
|---|---|---|---|
| Square | 95.2% (3914) | 80.5% (4148) | 93.0% (9637) |
| 0–1 | 67.6% (3914) | 84.9% (4148) | 95.7% (9637) |

a), it reflects well the poor *robust* generalization when tested with AWGN (Gilmer et al., 2019). Increasing the regularization coefficient $\lambda$ reduces ICB from 50% (a) to 23% (b) and 15% (c), and the robust GE from 38% (a) to 19% (b) and 10% (c). Both standard and robust GE are bounded at all times by ICB. The latter is, however, a coincidence, as the robust GE is subject to the arbitrary AWGN noise variance ($\sigma^2 = 1/16$). The additive noise variance could be increased to increase the robust GE beyond the range bounded by ICB. More important than bounding the robust GE *absolute percentage* is that ICB captures the trend of robust generalization. Evaluating robustness effectively is error-prone and often assumes access to test data and a differentiable model (Athalye et al., 2018; Carlini et al., 2019). We make no such assumptions here. The lack of robustness in Fig. 1 a) would have likely gone unnoticed. Next, we assess the extent to which ICB contains GEs evaluated on a variety of models and datasets.

### 4.1 Bounding generalization error

We refer to the percentage of cases where GE < ICB as the *"ICB satisfaction rate"*, or *"Sat."* in plots. We expect $\approx$ 95% of samples to satisfy this property as ICB is evaluated at 95% confidence ($\delta = 0.05$). We consider other $\delta$ values in §A.4. Overall ICB satisfaction rates are listed in Table 1 for two loss functions, and the ICB is plotted versus GE in Fig. 2 for the square loss. In general, ICB was satisfied at a high rate depending on the dataset. Next, we identify factors affecting the ICB satisfaction rate.

**Best-performing models** with test accuracy $\geq$ 70% attain an ICB satisfaction rate of 94.7% ($N = 792$) for `SVHN` and 0–1 loss. An 80% threshold for `CIFAR` led 94.0% ($N = 1438$) of samples to satisfy ICB.

**Architecture** type has a significant effect on ICB satisfaction rate: CNNs had an 11–14% greater ICB satisfaction rate than MLPs for `CIFAR`, and 10–18% greater rate for `SVHN`, where the range depended on the loss function. Intriguingly, the superior ICB satisfaction rate of CNNs was not due to greater test-set accuracy. CNNs had lower GE as a result of their lower *train*-set accuracy (Table A5).

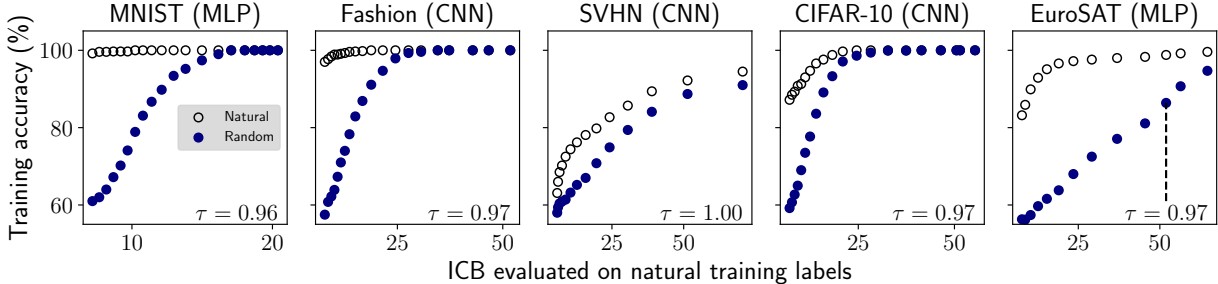

Figure 3: **The ICB often distinguishes between natural and randomized training sets.** Training accuracy for "Natural" and "Random" labels is plotted versus the theoretical GE bound (ICB), which is evaluated on the natural labels. Each data point corresponds to a unique regularization value $\lambda$, which influences the ICB value. We show the "0 vs. 1" task for all datasets, except CIFAR, for which the "7 vs. 8" task performed best (Table 2). Considerable separation between natural and randomly labeled sets is observed for all datasets. ICB is highly correlated with ability to fit random labels in all cases as shown by $\tau$ values. The broken vertical line for `EuroSAT` indicates the ICB value for which there is at least a 10% accuracy difference between natural versus random sets.

Table 2: **Ability of ICB to separate natural versus random training labels is a good predictor of ICB satisfaction rate by task.** The row "$ICB_{rand}@X\%$" indicates the minimum ICB value for which a $X\%$ accuracy difference between natural and random labels is observed. The "Sat. (%)" column showing the ICB satisfaction rate is taken from §4.1, Exp. A. The column $\tau$ indicates the rank correlation between $ICB_{rand}@X\%$ and Sat. (%) over the nine tasks. Columns sorted by ascending order of $ICB_{rand}@X\%$.

| | | | | | | | | | | | |
|---|---|---|---|---|---|---|---|---|---|---|---|
| EuroSAT | Task | 2/3 | 6/7 | 3/4 | 7/8 | 4/5 | 5/6 | 0/1 | 1/2 | 8/9 | |
| | Sat. % | 82 | 84 | 98 | 100 | 100 | 100 | 100 | 100 | 100 | $\tau = 0.76$ |
| | $ICB_{rand}@10\%$ | 7.5 | 10.0 | 11.1 | 13.2 | 18.5 | 21.6 | 41.6 | 47.9 | 51.2 | |
| CIFAR-10 | Task | 2/3 | 3/4 | 5/6 | 4/5 | 6/7 | 0/1 | 1/2 | 8/9 | 7/8 | |
| | Sat. % | 54 | 54 | 62 | 56 | 84 | 85 | 92 | 89 | 100 | $\tau = 0.87$ |
| | $ICB_{rand}@5\%$ | 7.4 | 9.4 | 11.8 | 12.1 | 12.5 | 12.6 | 14.8 | 15.2 | 17.8 | |

**Training time** GE generally increases with training, therefore the overall ICB satisfaction rate may be biased upward by the inclusion of models trained for a short time (e.g. $t = 10^2$). Selecting models trained for at least $t = 10^4$ steps, resulted in an ICB satisfaction rate *decrease* for `CIFAR-10` trained CNNs from 84.9% overall to 76.3% ($N = 2351$), and a slight *increase* for `SVHN` from 67.6% overall to 70.5% ($N = 2114$). Expressing GE in terms of MSE, ICB saturation rate fell from 95.2% overall to 91.3% ($N = 2114$).

**The number of training samples** impacted the ICB satisfaction rate for the 0–1 loss on `SVHN` (Fig. A7).

**Inter-task differences** in ICB satisfaction rate were observed (Table A6). For `EuroSAT`, six of nine tasks *always* satisfied ICB, and three tasks reduced the overall average. The satisfaction rate was 82.2% ($N = 1123$) for the "2 vs. 3" task and 83.5% ($N = 1154$) for the "6 vs. 7" task. For `CIFAR-10`, tasks "2 vs. 3" through "5 vs. 6" had ICB satisfaction rates considerably worse than the rest by a margin of $15 - 20\%$, and for `SVHN`, tasks "2 vs. 3", "5 vs. 6", and "8 vs. 9" were poor performing. These inter-task trends were consistent for the CNN and MLP architectures.

## 4.2 The randomization test

Zhang et al. (2017; 2021) proposed the "randomization test" after observing that DNNs easily fit random labels. They argued that generalization bounds ought to be able to distinguish models trained on *natural* versus *random* training labels, since generalization is by construction made impossible in the latter case. Motivated by the work of Zhang et al., we first hold all metaparameters constant and report ICB values

before and after randomizing the training labels (Table 3). We then regularized models until they could no longer fit *random* training labels, while still permitting them to fit the *natural* training labels (Table 2, Fig. 3). We consider only two-layer ReLU networks here, and consistent with §4.1, we use the CNN architecture for `FashionMNIST`, `SVHN` & `CIFAR` and an MLP for `MNIST` & `EuroSAT`. We train these models to $t = \infty$ on the natural training set ($N_{\text{trn}} = 1000$). Surprisingly, $\text{ICB}_{\text{UB}}$ approximates the GE well even when the model is trained on *random labels* (Table 3). For $\lambda = 0.1$, $\text{ICB}_{\text{UB}} = 15.5\%$ compared to a GE of 21.3%. Next, for $\lambda = 0.01$, $\text{ICB}_{\text{UB}} = 38.5\%$ and GE is 39.7%. Last, for $\lambda = 0.001$, $I_{\text{UB}} = 8.96$, which is greater than $\log(N_{\text{train}}) = 6.91$ nats, therefore the corresponding $\text{ICB}_{\text{UB}}$ of 197.6% should be discarded. In this case, substituting the "optimistic" lower estimate $\text{ICB}_{\text{LB}} = 54.1\% \approx \text{GE} = 50\%$.

We expect $I(X; Z)$ to be smaller after training on natural labels, since training on random labels requires memorization of random data, i.e., the opposite of compression. Importantly, to isolate the effect of the training label type, the training accuracy must be controlled, as higher accuracy generally requires greater complexity and thus larger $I(X; Z)$. This intuition is consistent with the results of Table 3, as both $I_{\text{LB}}$ and $I_{\text{UB}}$ increase monotonically with the training accuracy for both training label types. Training with $\lambda = 0.001$ allows models to perfectly fit both natural and randomized training sets (Table 3 rows with "Train" = 100%) and presents a suitable setting for evaluating the sensitivity of ICB to the training label type. Indeed, $I_{\text{LB}}$ is greater for random labels (6.37 vs. 5.78 nats), resulting in an increase of the *optimistic* theoretical GE bound, $\text{ICB}_{\text{LB}}$, from 40.5% to 54.1%. The more *pessimistic* $\text{ICB}_{\text{UB}}$ increases even more dramatically from 90.4% to 197.6%, which is beyond the valid range of GE ($0 - 100\%$).

Table 3: **ICB increases after training on random labels.** Randomization test results for `EuroSAT`. The lower and upper MI bounds, $I_{\text{LB}}$ and $I_{\text{UB}}$, are included for comparison against $\log(N_{\text{trn}}) \approx 6.91$ nats. Columns $\text{ICB}_{\text{LB}}$ and $\text{ICB}_{\text{UB}}$ refer to whether $I_{\text{LB}}$ or $I_{\text{UB}}$ is taken as $I(X; Z)$ estimate, respectively. Columns "Train" and "Test" show the respective accuracy in %. ICB values are larger for random labels when comparing rows with "Train"= 100.0.

| $\lambda$ | $I_{\text{LB}}$ | $I_{\text{UB}}$ | $\text{ICB}_{\text{LB}}$ | $\text{ICB}_{\text{UB}}$ | Train | Test | GE |
|---|---|---|---|---|---|---|---|
| Natural training labels | | | | | | | |
| $10^{-1}$ | 4.87 | 5.37 | 26.0 | 33.1 | 97.9 | 98.7 | -0.8 |
| $10^{-2}$ | 5.40 | 6.58 | 33.7 | 60.2 | 99.5 | 98.6 | 0.9 |
| $10^{-3}$ | 5.78 | 7.40 | 40.5 | 90.4 | 100.0 | 97.5 | 2.5 |
| Random training labels | | | | | | | |
| $10^{-1}$ | 3.68 | 3.75 | 15.0 | 15.5 | 71.3 | 50.0 | 21.3 |
| $10^{-2}$ | 5.28 | 5.67 | 31.7 | 38.5 | 89.7 | 50.0 | 39.7 |
| $10^{-3}$ | 6.37 | 8.96 | 54.1 | 197.6 | 100.0 | 50.0 | 50.0 |

**The randomization test identifies tasks with low ICB satisfaction** Recall from §4.1 that three binary classification tasks reduced the ICB satisfaction rate below 100% for `EuroSAT`: "2 vs. 3" (Sat. 82%), "6 vs. 7" (Sat. 84%), and "3 vs. 4" (Sat. 98%). We observed that these were the *same* tasks for which ICB performed poorly on the randomization test. Specifically, we measured the minimum ICB value for which an at least 10% accuracy difference was recorded between the natural and random training sets (vertical broken line in Fig. 3) when training with 20 different regularization values $\lambda$ in the range $10^{-4}$ to $10^1$. The "2 vs. 3" task required the smallest ICB value (7.5%) before this accuracy difference was reached between the two label types. The "6 vs. 7" task had the next highest ICB value of 10.0%, followed by "3 vs. 4" with 11.1%. The other six tasks—with 100% ICB satisfaction—had strictly greater ICB values (Table 2). Similar results are observed for `CIFAR-10` using a smaller 5% threshold as accuracies for natural and random labels were closer than for `EuroSAT`. The tasks with minimum ("2 vs. 3") and maximum ("7 vs. 8") satisfaction rate are the same tasks with the minimum and maximum $\text{ICB}_{\text{rand}}@5\%$. Therefore, the training-set based randomization test—which only required training a single model here—may be used to help identify when ICB performs well as a GE bound for a variety of models.

### 4.3 Vacuous or non-vacuous?

To evaluate whether ICB is close enough to GE to aid model comparison, we examined the model with the greatest accuracy on the test set with AWGN for each `SVHN` and `CIFAR` task. We used AWGN accuracy rather than standard test accuracy to select these models given the observation from Figure 1 that ICB may be more

Table 4: **The ICB is close enough to the GE for model comparison.** For five `SVHN` and `CIFAR` classification tasks (beginning with even numbers), we select the model with maximum accuracy on the test set with AWGN, shown in the "Robust" column. ICB is compared to the standard GE as well as the Robust GE ("RGE" column). High train-set accuracy ("Train" $\geq$ 99%) tends to coincide with large RGE and ICB values, making ICB loose w.r.t. GE. Otherwise, ICB is consistently close to GE (compare values in **bold**).

| | CIFAR | | | | | | SVHN | | | | | |
|---|---|---|---|---|---|---|---|---|---|---|---|---|
| Task | Train | Test | Robust | GE | RGE | ICB | Train | Test | Robust | GE | RGE | ICB |
| 0/1 | 96 | 86 | 83 | **10** | 12 | **13** | 87 | 79 | 68 | **8** | 19 | **15** |
| 2/3 | 93 | 74 | 71 | **18** | 21 | **25** | 100 | 85 | 67 | 15 | 32 | 46 |
| 4/5 | 99 | 77 | 73 | 22 | 26 | 52 | 100 | 88 | 70 | 12 | 29 | 67 |
| 6/7 | 93 | 85 | 83 | **8** | 10 | **12** | 90 | 78 | 70 | **12** | 20 | **15** |
| 8/9 | 85 | 80 | 80 | **5** | 5 | **7** | 99 | 77 | 66 | 23 | 33 | 62 |

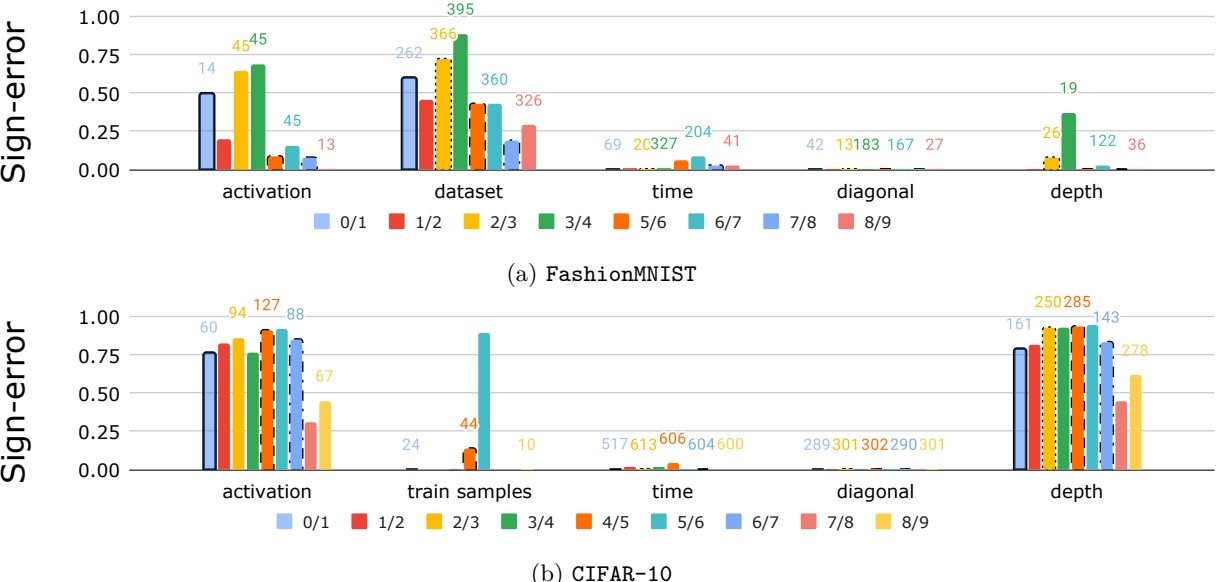

(a) `FashionMNIST`

(b) `CIFAR-10`

Figure 4: **The ability of ICB to predict GE varies for different metaparameters.** Plots show prediction Sign-Errors (SEs) (where *lower* is better) for robust GE with AWGN for six different metaparameter interventions on nine binary classification tasks $\{0 \text{ vs. } 1, 1 \text{ vs. } 2, \ldots, 8 \text{ vs. } 9\}$ for a) `FashionMNIST` and b) `CIFAR-10`. Numbers above the SE for even-numbered tasks indicate the number of samples $N$ comprising the respective SE measurement. The `train samples` category had insufficient $N$ for `FashionMNIST` and was therefore replaced with a `dataset` category in a), which examines the effect of switching the dataset from `FashionMNIST` to `CIFAR-10`. The `dataset` category is omitted in b) as SE is symmetric. Note that mean SE is equal to max SE for the `activation` metaparameter which had only one combination (ReLU versus Erf). See Table A7 for a detailed example showing how a column is computed in these plots.

aligned with *robust* GE (also see Fig. A8). The most accurate models on the standard test set often recorded zero error on the training set and thus incurred large ICB values. Here, ICB values were considerably less than 50% in all cases except where training accuracy was 99% or greater (Table 4). Furthermore, ICB values were often close to the empirical GE, e.g., an ICB of 7% was obtained for GE = 5% for the `CIFAR` task `8/9` using only 1500 training samples (Table 4).

### 4.4 Effect of metaparameters on theoretical bound

The goal of this section is to assess the ability of ICB to robustly rank GEs for a variety of models trained by different metaparameters. We perform *coupled-network* experiments as per Dziugaite et al. (2020) to assess the ability of ICB to predict whether GE *increases* or *decreases* for all metaparameter combinations when manipulating one held-out metaparameter at a time. Ranking is assessed in terms of Sign-Error (SE):

$$\text{SE}(P^e, \text{ICB}) = \frac{1}{2} \mathbb{E}_{(w,w')\sim P^e} \left[ 1 - \text{sgn} \left( \text{GE}\left(w'\right) - \text{GE}\left(w\right) \right) \cdot \text{sgn} \left( \text{ICB}\left(w'\right) - \text{ICB}\left(w\right) \right) \right]. \tag{5}$$

The SE (Eq. 5) is evaluated with respect to different assignments of the metaparameters, $w$ and $w'$, which are said to be drawn from a distribution $P^e$ induced by environment $e$, and differ in only one metaparameter value at a time. Note that in Dziugaite et al. the expectation in Eq. (5) is taken over randomly initialized finite DNN parameters and SGD mini-batch ordering. Conversely, infinite-ensembles trained by GD are deterministic. Therefore, the expectation in Eq. (5) is with respect to the choice of metaparameters and random training sample only.

We include three metaparameters that were not present in the work of Dziugaite et al.: the `diagonal` regularization, training `time`, and `activation` function. We omit `width`, `learning rate`, and `batch size`, as they do not apply to our setting. We report the mean SE over all possible interventions to each metaparameter in Fig. 4. The maximum or "max" sign-error advocated by Dziugaite et al. is discussed in text, as it was subject to greater noise than mean SE due to the smaller sample size.[4] To calculate the mean SE for the "depth" metaparameter, for example, which has a range of 1 to 5, we evaluate the SE arising from changing the depth from 1 to 2, again for 1 to 3, and so on for all ten combinations. The SE is then averaged across the ten "before" and "after" configurations, whereas the max SE refers to the one combination with the greatest SE, e.g., changing the depth from 4 to 5. We discard samples with a Hoeffding weight less than 0.5 as per Dziugaite et al., indicating that a GE difference is too small relative to the number of train-set and test-set samples to reliably measure changes in the *true error* rate (i.e., on the underlying data-generating process).[5] We discard cases with $N < 10$ for the max SE, whereas we compute a sample-weighted average for the mean SE based on all cases. Few measurements of the *standard* GE satisfied the Hoeffding threshold for `FashionMNIST`, therefore we added AWGN to the test set to increase the typical GE and number of valid cases to analyze. We repeat our analysis for standard GE for `CIFAR-10` in Fig. A9 and contrast it with the AWGN case.

The `diagonal` regularization was the most consistent metaparameter, with zero max SE for all tasks and datasets. Intervening on the training `time` consistently led to small mean and max SE, as well as for the number of `train samples` for the majority of tasks (Fig. 4). The `time` metaparameter had a max SE of 0.45 for the 6/7 `FashionMNIST` task—consistent with the peak mean SE for task 6/7 in Fig. 4a—and five tasks had a max SE of zero. For `CIFAR-10`, max SE w.r.t. `time` was 0.27, with five tasks having max SE $< 0.02$. The max SE for `train samples` was 0.89 ($N = 19$) for task 5/6, which was followed by 0.14 ($N = 44$) for task 4/5. Other tasks either had zero max SE or insufficient data. Note that these `CIFAR-10` tasks 4/5 and 5/6 also had adverse performance for the randomization test (Table 2). In summary, the three metaparameters: `diagonal`, `time`, and `train samples` strongly influence overall correlation between ICB and GE.

Manipulating the `activation` function and `depth` resulted in a greater discrepancy between `FashionMNIST` and `CIFAR-10`. For these metaparameters ICB was considerably more predictive for the former dataset resulting in a lower SE. For example, for the `depth` metaparameter the max SE for `FashionMNIST` was 0.23 followed by 0.06, compared to 1.0 for `CIFAR-10`. Results for `EuroSAT` were closer to `FashionMNIST` than `CIFAR-10` and may be found in Fig. A10.

---

[4]We use the term "max SE" rather than "robust SE" from Dziugaite et al. to avoid confusion with the "robust GE". The latter is evaluated on a test set with AWGN and is a different concept from the former.

[5]Recall from the Introduction that we merely *operationalized* GE as the difference between train-set accuracy and the test-set accuracy. In contrast, the GE is formally defined as the difference between train-set accuracy and the *true* accuracy (or error); test-set accuracy is an approximation for the latter, which we account for with this step.

## 5 Discussion

Our results show that the ICB serves as a non-vacuous generalization bound, which we verified in the case of infinite-width networks. Furthermore, we performed a broader evaluation than is typically considered for theoretical GE bounds: i) We searched for ICB violations by evaluating ICB throughout training, rather than at a specific number of epochs or training loss value. ii) We varied the number of training samples and classification labels, compared to a static train/test split. iii) We considered robust GE in addition to standard GE. iv) Experiments were performed on five datasets.

ICB was satisfied at around the 95% rate for four of five datasets when GE was expressed in terms of MSE, or three of five when GE was expressed in terms of accuracy. Best-performing models with at least $70 - 80\%$ test accuracy consistently satisfied ICB for all datasets, which is encouraging, since accurate models are more likely to be deployed. The `SVHN` dataset had the lowest ICB satisfaction rate among all cases for the 0–1 loss, however, this may have been impacted by too few training examples. Furthermore, we identified that the training label randomization test may detect specific tasks with low ICB satisfaction rate; ensuring ICB yields different results for natural and randomized training labels may greatly improve results. CNNs had superior performance compared to MLPs. Although the MRF assumption does not appear to be necessary to derive ICB, the success of CNNs in comparison to MLPs may be related to the MRF assumption and 2D images having only local correlations, compared to their flattened counterparts used with MLPs (see §A.1).

The ICB was able to successfully predict changes in robust GE when the digonal regularization, training time, and number of training samples varied. ICB was less successful when the depth and activation function varied for `CIFAR-10`, yet performed well for `FashionMNIST` as well as for `EuroSAT`. The precise NTK limit we consider is generally assumed to only hold for shallow models, which may explain some of the failure cases for the depth metaparameter (Li et al., 2021). It is also worth considering to what extent a generalization bound *ought* to be able to rank GEs, given that it is by definition merely an upper bound on the error. For example, GEs of 1% and 29% are both compatible with a bound of 30%, which would contribute to poor GE prediction on an absolute scale.

**Relevance to deep learning** One should use caution before extrapolating our conclusions based on infinite-width networks to finite-width DNNs. The ability of infinite-width networks to approximate their finite-width counterparts is reduced with increasing training samples (Lee et al., 2019), regularization (Lee et al., 2020), and depth (Li et al., 2021). Nevertheless, the infinite-width framework has allowed us to demonstrate the practical relevance of the ICB for an exciting family of models as a first step. It has been argued that understanding generalization for shallow kernel learning models is essential to understand generalization behaviour of deep networks. Kernel learning and DL share the ability to exactly fit their training sets yet still generalize well, a phenomenon that other bounds fail to explain (Belkin et al., 2018). We leave the study of ICB for finite-width DNNs to future work, which may require alternative MI estimation techniques.

## 6 Related Work

**Kernel-regression generalization error** Canatar et al. (2021b) derived an analytical expression for the generalization MSE of kernel regression models using a replica method from statistical mechanics. Their predictions show excellent agreement with the empirical GE of NTK models on `MNIST` and `CIFAR` datasets as a function of the training sample size. Furthermore, their method is sensitive to differences in difficulty between similar classification tasks, e.g., showing that `MNIST` "0 vs. 1" digit classification is easier to learn than "8 vs. 9". Canatar et al. (2021a) extend the method to predict out-of-distribution GE. An alternative method is the Leave-One-Out (LOO) error estimator (Lachenbruch, 1967). LOO is generally impractical for DL due to the computational requirement of training $N$ DNNs on $N$ different training sets. However, Bachmann et al. (2022) proposed a closed-form LOO estimator based on a kernel regression model trained on the complete training set once. Their estimator shows excellent agreement with test MSE and accuracy for a five-layer ReLU NTK model trained on `MNIST` and `CIFAR`. While Bachmann et al. averaged results over five training sets of size $500 - 20000$, we only draw a single training set of $250 - 2000$ samples for each set of metaparameters. Our choice was made to reflect a practical "small data" scenario, where GE has to be bounded using a modest set of labeled data. As a result, however, our GE and ICB estimates have

greater variance than those of Bachmann et al. We used the infinite-width DNN limit for convenience and as a first step to assess the efficacy of ICB; we did not set out to find optimal generalization bounds for kernel regression. An advantage of ICB is that it only requires access to $I(X; Z)$—a black-box statistic applicable to a wide variety of models beyond kernel regression. Therefore, ICB may become increasingly relevant for DLs using MI estimators with different strengths and assumptions, e.g., with distributional constraints on weight matrices (Gabrié et al., 2018) or infinite-depth corrections (Li et al., 2021).

**Generalization bounds for deep learning** Dziugaite & Roy (2017) develop a PAC-Bayes GE bound and evaluated it on a `MNIST` binary classification task using the complete training set ($N_{\text{train}} = 55\,\text{k}$) and a fully-connected NN with 2-3 layers and ReLU activations. Although their bound was non-vacuous ($\approx 20\%$), it was several times larger than the error estimated on held-out data ($< 1\%$). A comparison with our work is difficult, as we did not use finite-width DNNs. We showed that the ICB yields a smaller ($\approx 10\%$) bound from less than 2000 samples for several classification tasks. Zhou et al. (2019) proposed a PAC-Bayes generalization bound based on the compressed size of a DNN after pruning and quantization. They obtain a GE bound of 46% for MNIST and $96 - 98\%$ for ImageNet. The measure of compression used by Zhou et al. (2019) is distinct from input compression in terms of MI here. The bounds of Dziugaite & Roy and Zhou et al. concern model complexity, whereas ICB is based on data compression by the hidden layers. Both Dziugaite & Roy and Zhou et al. optimized their bounds for best results, whereas we used standard training procedures.

**Generalization bounds from unlabeled data** GE bounds or estimates may be obtained without directly estimating model complexity. Garg et al. (2021) leverage the so-called "early learning" phenomenon, whereby DNNs fit true labels before noisy labels, to develop a post-hoc GE bound. They validate their bound on NTK-based wide DNNs, CNNs, and LSTMs. In contrast to our work, the Garg et al. bound requires additional unlabeled data, that in practice, can be carved out from the training set. They assign random labels to the carved-out set, and augment the training set with this random data. Their bound is based on the empirical error computed on both the clean and random set. Empirically, Garg et al. (2021) show that it may be possible to maintain model accuracy when training on partially randomized labels in some settings by using weight decay or early stopping. Unfortunately, random labels reduce the task signal-to-noise ratio, $I(X; Y)$, and may be challenging to apply with unregularized models that nonetheless generalize well (Zhang et al., 2017). Jiang et al. (2022) observed that the disagreement of separately trained DNNs on *unlabeled* held-out datasets is similar to the disagreement of those models on a *labeled* held-out set. Their claim follows an empirical observation that deep ensembles are often well-calibrated, however, this calibration property may not always hold in important settings (Kirsch & Gal, 2022).

**Information compression and generalization** The MI $I(S; w)$ between the training data $S = (x, y)$ supplied as input to a stochastic learning algorithm and the weights $w$ it outputs can also serve to bound GE (Xu & Raginsky, 2017; Achille & Soatto, 2018). Decomposing $I(S; w)$ into $I(w; x) + I(w; y|x)$, Harutyunyan et al. (2020) show that reducing the second term—the information $w$ contain about the labels $y$ beyond what can be inferred from $x$—is key to avoid unintended memorization. As a result, these works *optimize* MI bounds, whereas we seek to *measure* MI to evaluate a GE bound. Furthermore, Shwartz-Ziv & Alemi (2020)[Appendix C.7] evaluated $I(S; w)$ for infinite-width networks and found that it tends to infinity as the training time goes to infinity. Thus, a GE bound based on $I(S; w)$ is vacuous for these networks which nevertheless generalize well. Saxe et al. (2018) observed a lack of compression in ReLU networks and argued that compression must be unrelated to generalization in DNNs, since it is known that ReLU networks generalize well. However, their binning procedure based on Paninski (2003) involves metaparameters that influence entropy and MI estimation. Other works have studied input compression in linear regression (Chechik et al., 2005) and finite-width ReLU DNNs using adaptive binning estimators (Chelombiev et al., 2019). We use MI bounds free from such metaparameters and observe input compression regardless of the nonlinearity type, consistent with Shwartz-Ziv & Alemi (2020). We are excited about future work on input compression phenomena and the challenging case of finite-width DNNs.

## 7 Conclusion

We assessed the ICB along three performance axes: tightness, percentage of trials satisfying the bound, and correlation with GE. Empirical results show that input compression serves as a simple and effective generalization bound, complementing previous theory. Additionally, ICB can help pinpoint interesting failures of robust generalization that go undetected by standard generalization metrics. An important consequence of the ICB with respect to NAS is that *bigger is not necessarily better*, at least in terms of the information complexity of infinite-width networks. Equally important as the architecture are the metaparameters and training duration, all of which affect input compression. Consistent with Occam's razor, less information complexity—or more input compression—yields more performant models, reducing the upper bound on generalization error. We conclude that input compression, which is data-centric, is a more effective complexity metric than model-centric proxies like the number of parameters or depth.

### Author Contributions

Angus Galloway devised the study and wrote the first draft in consultation with all co-authors. Anna Golubeva critically revised the entire manuscript and made considerable contributions to the theoretical background. Mahmoud Salem provided experiment support with adversarial attacks in the JAX framework, and characterizing the finite to infinite-width NN correspondence. These experiments informed the final scope of the study and addressed alternate hypotheses. Mihai Nica, Yani Ioannou, and Graham W. Taylor provided technical advice and revised the manuscript. All authors consented to submission of this work to TMLR.

### Acknowledgments

This research was developed with funding from the Defense Advanced Research Projects Agency (DARPA). The views, opinions and/or findings expressed are those of the author and should not be interpreted as representing the official views or policies of the Department of Defense or the U.S. Government. Graham W. Taylor and Angus Galloway also acknowledge support from CIFAR and the Canada Foundation for Innovation. Angus Galloway also acknowledges supervision by Medhat Moussa. Resources used in preparing this research were provided, in part, by the Province of Ontario, the Government of Canada through CIFAR, and companies sponsoring the Vector Institute: `http://www.vectorinstitute.ai/#partners`. Anna Golubeva is supported by the National Science Foundation under Cooperative Agreement PHY-2019786 (The NSF AI Institute for Artificial Intelligence and Fundamental Interactions, `http://iaifi.org/`). Mihai Nica is supported by an NSERC Discovery Grant.

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

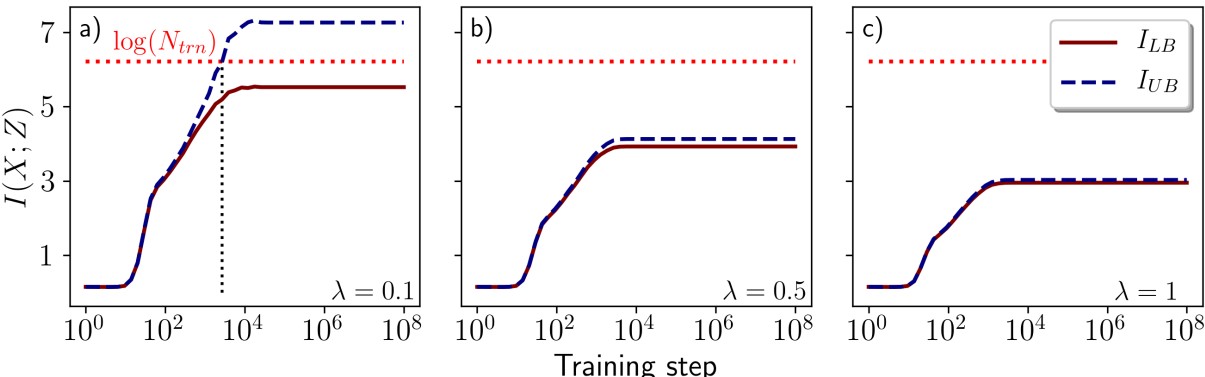

Figure 5: We plot $I(X;Z)$ upper (2) and lower (6) bounds corresponding to the illustrative `EuroSAT` example (Figure 1). Increasing the regularization to $\lambda = 0.5$ in b) and $\lambda = 1.0$ in c) reduces MI below $\log(N_{\text{trn}})$. Samples to the right of the vertical line in a) where $I_{UB}$ crosses $\log(N_{\text{trn}})$ are discarded for the main analyses. NB: We use natural units ("Nats" or "Shannons") for $I(X;Z)$ here, but we convert to bits when evaluating the ICB.

## A    Appendix

### A.1    Assumptions of input compression bound

It is assumed in the construction of the ICB that $X$ is a $d$-dimensional random variable that obeys an ergodic MRF probability distribution, asymptotically in $d$. A MRF is an undirected graphical model, used to model data distributions with a particular conditional independency structure, which is commonly used for spatial data, including images (see (Murphy, 2012)[Ch. 19] for an introduction). Mathematically, this means that $p(x)$ factorizes into a product of terms which represent the potentials for each clique on the underlying graph. In terms of correlations, this means that each pixel in a 2D image is strongly correlated with its immediate neighbors, but not with pixels that are further away. The "ergodic" part is essential for the derivation of the ICB: An ergodic MRF does not "get stuck" in any part of the state space; in other words, there is a nonzero probability for every possible state to be reached. Ultimately, this is the necessary assumption to invoke the Asymptotic Equipartition Property (AEP), which in turn allows us to invoke typicality. Defining the typical set is the crux of the ICB derivation, because it enables us to quantify the hypothesis space cardinality in terms of entropy. From here, the rest follows from information-theory fundamentals.

### A.2    Lower bound on MI

We may lower bound $I(X;Z)$ using a bound of similar form as equation 2 based on a batch of $N$ samples:

$$I(X;Z) \geq \mathbb{E}\left[\frac{1}{N}\sum_{i=1}^{N}\log\frac{p(z_i|x_i)}{\frac{1}{N}\sum_j p(z_i|x_j)}\right] = I_{\text{LB}}\,, \tag{6}$$

where the expectation is taken over $N$ independent samples from the joint distribution $\prod_j p(x_j, z_j)$. The main difference between this bound and equation 6 is the inclusion of $p(z_i|x_i)$ in the denominator.

### A.3    Illustrative example and filtering MI

We empirically verified that equation 6 and equation 2 yield similar results when $I_{\text{UB}} < \log(N_{\text{trn}})$ (Fig. A5).

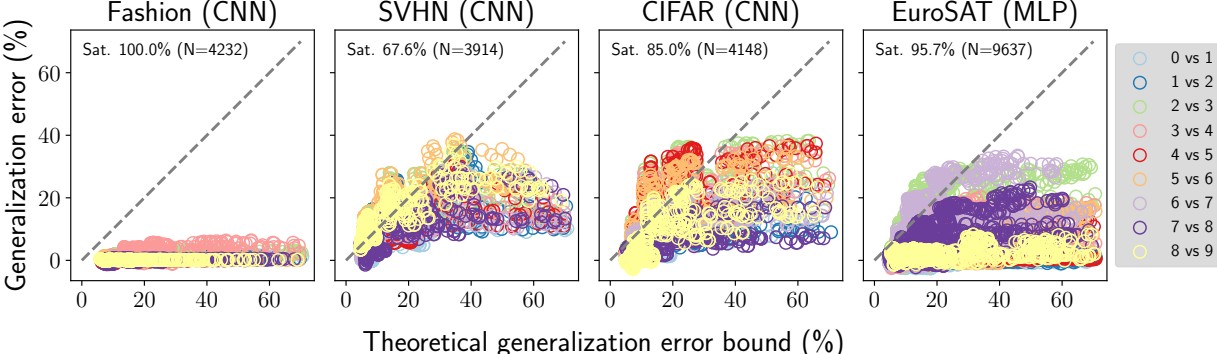

Figure 6: ICB is plotted versus GE (0–1 loss) for `FashionMNIST`, `SVHN`, `CIFAR-10`, and `EuroSAT` datasets. The ICB satisfaction rate is annotated in the top left corner of each plot with format "ICB % (N)". Each binary classification task is assigned a unique colour to highlight inter-task differences in ICB satisfaction rate. See Figure 2 of §4.1 for the corresponding Figure with GE expressed using square loss. NB: Results for `MNIST` omitted from Figure as they were similar to `FashionMNIST`.

Table 5: CNNs have consistently smaller average GE than MLPs for `SVHN` and `CIFAR-10`, as they do not fit the training set to the same extent as MLPs. Note that these average accuracies include models trained for both short and long periods of time, therefore a low test accuracy here does not imply failure of learning.

| Dataset | Arch. | ICB Sat (Acc./MSE) (%) | Train (%) | Test (%) | GE (%) |
|---------|-------|------------------------|-----------|----------|--------|
| SVHN | CNN | 67.6%/95.7% (3914) | 74.78 | 64.18 | 10.60 |
| | MLP | 49.5%/86.2% (3980) | 79.82 | 67.07 | 12.74 |
| CIFAR-10 | CNN | 84.9%/80.5% (4153) | 85.84 | 76.12 | 9.72 |
| | MLP | 73.8%/68.0% (4318) | 88.13 | 76.70 | 11.43 |

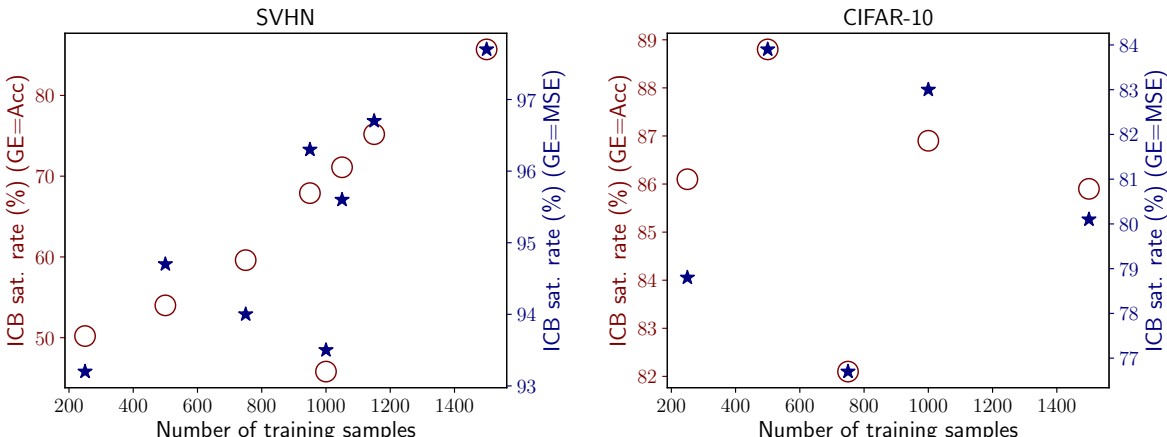

Figure 7: The ICB satisfaction rate is plotted versus the number of training samples for `SVHN` and `CIFAR-10` datasets. The primary vertical axis bounds GE using the 0–1 loss (GE=Acc), whereas the secondary vertical axis bounds GE using the square loss (GE=MSE). Here, we only consider the first three tasks for each dataset, i.e., "0 vs 1", "1 vs. 2", and "2 vs. 3" as an approximation for all nine tasks. Other datasets `FashionMNIST` and `EuroSAT` were excluded as they consistently satisfied ICB for all train-set sizes considered here.

### A.4 Bounding generalization error

**Loss function** We measured GE using the 0–1 and square loss (see Fig. A6 for 0–1 loss). This change results in no difference in the overall ICB satisfaction rate for `FashionMNIST`, a decrease for `SVHN` from 95.7% to 67.6%, a small increase for `CIFAR-10` from to 80.5% to 84.9% as well as for `EuroSAT` from 93.6% to 95.7%.

**Train-set size** The number of training samples may have affected the ICB satisfaction rate for some datasets (Fig. A7). For this experiment, we included small training sets of size $N_{\text{trn}} \in \{250, 500, 750\}$ to observe a broader relationship between $N_{\text{trn}}$ and the ICB satisfaction rate. For `SVHN`, the ICB satisfaction rate appears to be still improving at $N = 1500$ despite a dip at $N = 1000$. We ran additional experiments with $N_{\text{trn}} \in \{950, 1050, 1150\}$ to assess the width of this dip for `SVHN`. The dip at $N = 1000$ is not necessarily alarming as other studies have observed that more training data can sometimes hurt model performance, e.g., see Nakkiran et al. (2019). The ICB satisfaction rate for `SVHN` may therefore benefit from additional training data given the overall positive trend. Conversely, there is no discernible trend for `CIFAR-10`.

**Task labels** There were considerable inter-task differences in ICB satisfaction rate (Table A6). The "Overall" row of Table A6 corresponds to the ICB satisfaction rates presented in Table 1, as well as in the top left corners of Fig. 2 and Fig. A6 for the square and 0–1 loss respectively.

**Confidence** The confidence parameter $\delta$ in the PAC framework accounts for a small probability that the training set received by a learning algorithm is misleading, i.e., the samples are not representative of the true data-generating distribution, or they do not reflect all relevant details of the distribution (Shalev-Shwartz & Ben-David, 2014). It is natural to wonder how the choice of $\delta$ affects the ICB satisfaction rate. Recall from §4.1 that we refer to the percentage of tuples (ICB, GE) for which GE < ICB as the "ICB satisfaction rate", which should roughly equal $1 - \delta$ expressed as a percentage, e.g., $\delta = 0.05$ equals 95% confidence. Decreasing $\delta$ to 0.01 from its default value of 0.05 increased the ICB satisfaction rate by 6.2% for `SVHN` on the 0–1 loss, compared to an expected $\approx 4\%$ difference (99% − 95%). Increasing $\delta$ to 0.20 decreased the ICB satisfaction rate by 11.2% compared to an anticipated difference of $\approx 19\%$ (95% − 80%). Results were less sensitive to $\delta$ for `CIFAR-10` and `EuroSAT` ($\approx 1\%$ difference for 99% − 95%), and independent from $\delta$ for `FashionMNIST` as the compression term alone (i.e. from the numerator of equation 4) sufficed to bound the small GEs for this dataset.

Table 6: ICB satisfaction rates broken down by task for three datasets. `MNIST` and `FashionMNIST` are excluded for brevity as they were always at 100%. Columns "Acc." and "MSE" indicate whether GE is quantified in terms of classification accuracy (0–1 loss) or MSE (square loss) respectively. The "N" column indicates the number of valid experiments.

| Task | SVHN CNN | | | CIFAR CNN | | | EuroSAT MLP | | |
|---|---|---|---|---|---|---|---|---|---|
| | Acc. | MSE | N | Acc. | MSE | N | Acc. | MSE | N |
| 0 vs 1 | 87.7% | 99.3% | 431 | 93.3% | 86.6% | 461 | 100.0% | 100.0% | 1012 |
| 1 vs 2 | 60.2% | 93.6% | 435 | 95.7% | 88.7% | 462 | 100.0% | 100.0% | 961 |
| 2 vs 3 | 51.8% | 94.1% | 440 | 69.9% | 69.2% | 468 | 82.2% | 82.2% | 1123 |
| 3 vs 4 | 78.7% | 97.2% | 431 | 74.5% | 71.0% | 459 | 97.9% | 86.0% | 1175 |
| 4 vs 5 | 94.2% | 97.0% | 434 | 71.3% | 71.3% | 460 | 100.0% | 100.0% | 1151 |
| 5 vs 6 | 30.0% | 88.6% | 440 | 74.8% | 72.8% | 460 | 100.0% | 100.0% | 1066 |
| 6 vs 7 | 85.7% | 97.0% | 433 | 91.6% | 83.2% | 463 | 83.5% | 78.3% | 1154 |
| 7 vs 8 | 84.4% | 97.7% | 430 | 100.0% | 94.1% | 461 | 100.0% | 94.9% | 1135 |
| 8 vs 9 | 37.1% | 92.7% | 440 | 93.5% | 87.4% | 459 | 100.0% | 100.0% | 860 |
| Overall | 67.8% | 95.2% | 3914 | 84.9% | 80.5% | 4153 | 96.0% | 93.5% | 9637 |

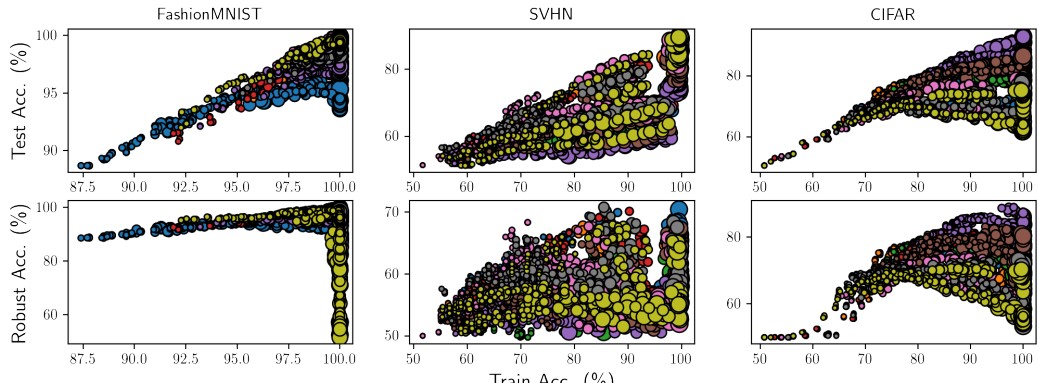

Figure 8: Train-set accuracy is plotted versus clean test accuracy (top) and robust test accuracy (bottom) using AWGN. The marker size indicates the ICB value, and each binary classification task is assigned a unique colour.

### A.5 Vacuous or non-vacuous?

We plotted train-set accuracy versus test accuracy on the "clean" and "noisy" sets, indicating ICB values by the marker size. Large ICB values tend to occur for very high training accuracy and coincide with poor robust accuracy (Fig. A8). Since ICB increased considerably for the models with highest clean test accuracy, we selected models with greatest robust accuracy in §4.3. These results are consistent with previously observed accuracy versus robustness trade-offs (Tsipras et al., 2019).

## B Effect of metaparameters on theoretical bound

We repeat the analysis from §4.4 for the `CIFAR-10` dataset in Fig. A9, this time comparing SE results when GE is measured using the standard test set versus a test set with AWGN. Evaluating GE with a noisy test set yields greater a sample size after filtering out measurements where GE is too small to reliably measure a change in a classifiers true error. The SE values for the `activation` and `diagonal` metaparameters are also slightly larger for the standard GE, implying that ICB is better correlated with robust GE than standard GE in our experiments for `CIFAR-10`. A detailed worked example showing how a column is computed in these

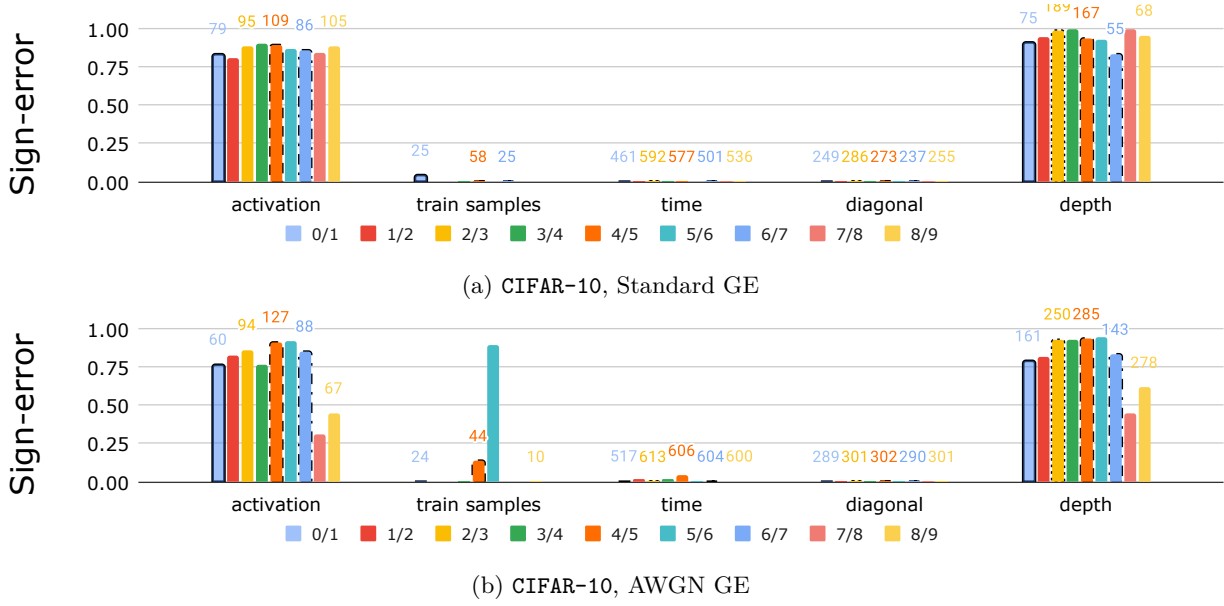

(a) `CIFAR-10`, Standard GE

(b) `CIFAR-10`, AWGN GE

Figure 9: **Evaluating GE on a noisy test set yields greater sample size after accounting for Monte Carlo variance in GE estimates, and slightly smaller sign-errors.** Plots show GE prediction SEs (where *lower* is better) for six different metaparameter interventions on nine binary classification tasks $\{0 \text{ vs. } 1, 1 \text{ vs. } 2, \ldots, 8 \text{ vs. } 9\}$ for `CIFAR-10`. Numbers above the SE for even-numbered tasks indicate the number of samples $N$ comprising the respective SE measurement. See Table A7 for a worked example showing how a column is computed in these plots.

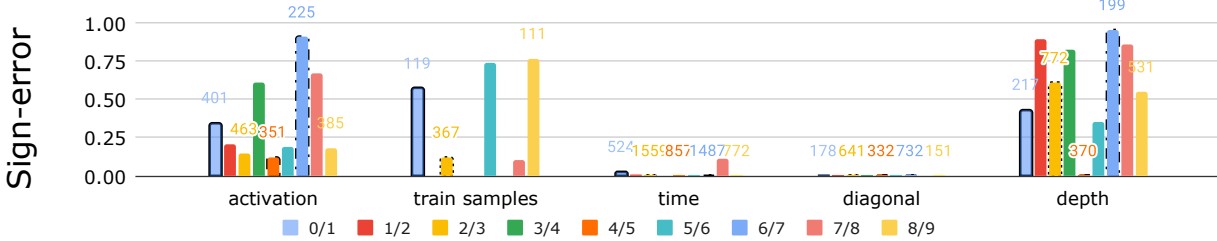

Figure 10: **Evaluating SE for `EuroSAT` with AWGN test set.** Other details similar to those described in captions for Fig 4 and Fig A9.

plots is provided in Table A7. Smaller SEs are observed on average for the `EuroSAT` dataset than for `CIFAR-10`, but with greater inter-task variance (Fig. A10). The SE is consistently near zero for the `time` and `diagonal` metaparameters across all datasets.

Table 7: Detailed GE prediction SE for all combinations of the `depth` metaparameter for the `CIFAR` AWGN 8 versus 9 task. The first two columns ("Raw") comprise all SEs with only basic filtering for $I_{\mathrm{UB}}(X; Z) \leq$ log(`train samples`). The next set of ("Filtered") columns additionally accounts for Monte Carlo variance of empirical averages and discards samples with a small difference in GE relative to the number of train and test samples (see text for details). The final "filtered" weighted average SE of 61.5% ($N = 278$) appears in Fig. A9b (series `8/9`).

| Raw | | Filtered | | | Depth | |
|---|---|---|---|---|---|---|
| **N** | **Sign-error** | **N** | **Sign-error** | **Weighting** | **Before** | **After** |
| 94 | 46.8% | 34 | 35.3% | 12 | 1 | 2 |
| 93 | 49.5% | 42 | 45.2% | 19 | 1 | 3 |
| 91 | 51.6% | 55 | 49.1% | 27 | 1 | 4 |
| 87 | 54.0% | 60 | 51.7% | 31 | 1 | 5 |
| 93 | 49.5% | 4 | 100.0% | 4 | 2 | 3 |
| 91 | 62.6% | 22 | 100.0% | 22 | 2 | 4 |
| 87 | 83.9% | 32 | 87.5% | 28 | 2 | 5 |
| 91 | 73.6% | 1 | 100.0% | 1 | 3 | 4 |
| 87 | 90.8% | 26 | 96.2% | 25 | 3 | 5 |
| 84 | 95.2% | 2 | 100.0% | 2 | 4 | 5 |
| | | 278 | 61.5% | | | |

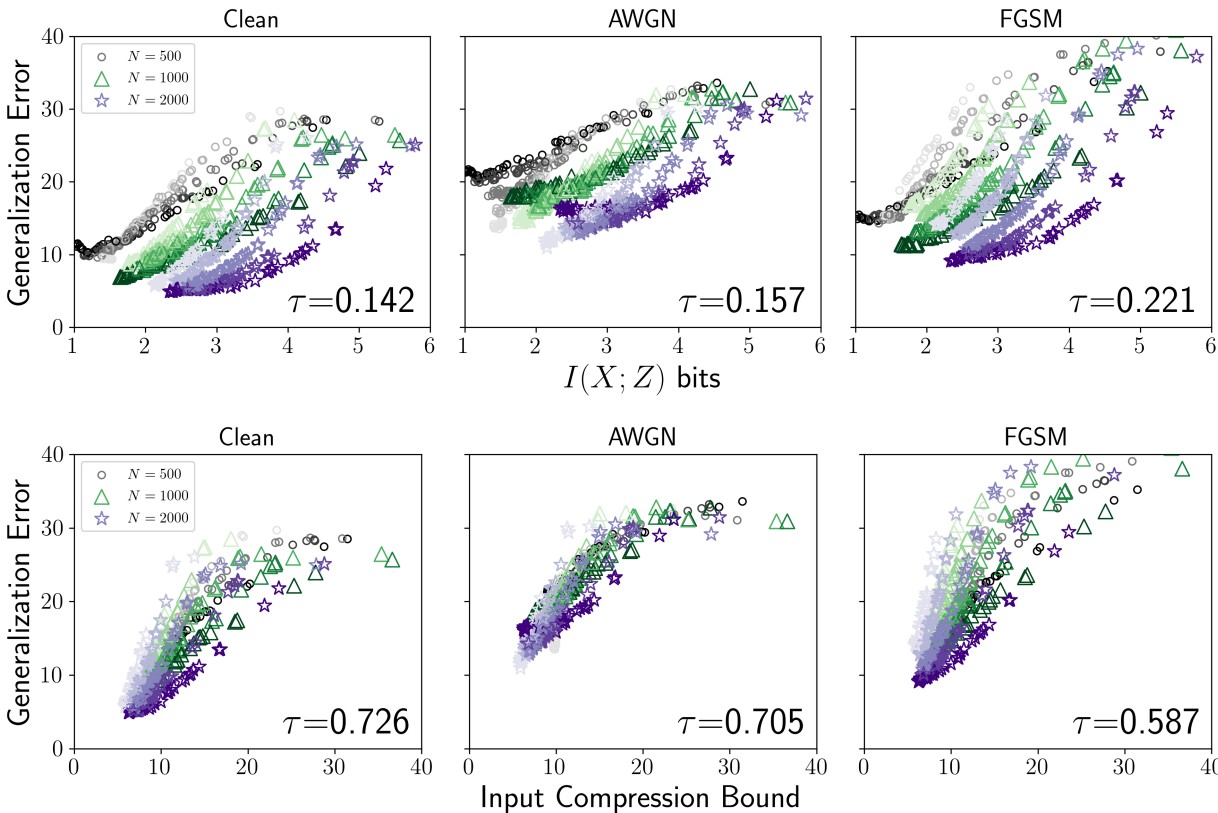

Figure 11: **ICB (bottom) ranks GEs better than $I(X;Z)$ alone (top) for different training set sizes.** Shown are 750 fully-connected NTK ReLU models trained ($t = \infty$) on a `CIFAR-10` binary classification task (classes 2 and 5) using three different training set sizes of $N = \{500, 1000, 2000\}$ and a test set with $N = 2000$. Model depth is indicated by the colour intensity for each series, where the darkest shade indicates the maximum depth of five (5) layers. Three GE types are evaluated: `Clean` (standard), AWGN (adversarial), and Fast Gradient-Sign Method (FGSM) (adversarial) are plotted with respect to $I(X;Z)$ (*top row*) and ICB (*bottom row*). Plotting GE versus the ICB better aligns results for different sized training sets ($N$) compared to $I(X;Z)$, and yields a better ranking in terms of Kendall-$\tau$.

### B.1 Advantage of ICB versus MI

To gain further insight into ICB, we examine GEs for a specific `CIFAR-10` binary classification task (classes 2 and 5) using three different training set sizes. Plotting GEs with respect to $I(X;Z)$ alone yields a poor overall ranking, whereas ICB effectively aligns trials with different training set sizes (Figure 11).

