# OpenReview forum: "Bounding generalization error with input compression: An empirical study with infinite-width networks"
_TMLR — Accepted by TMLR_

### Review · Reviewer_g7Gf · 2022-08-04

**Summary Of Contributions:**


This paper performs an empirical study of an information bottleneck-inspired generalization bound, ICB. The evaluation of the bound leverages relatively recent empirical approaches to measuring the mutual information, and the work applies this methodology to answer a range of questions about the ICB in an infinite-width model of deep neural networks. First, the paper attempts to falsify the bound on real-world data, succeeding on some datasets and suggesting that the assumptions required in the derivation of the ICB do not hold in many real-world datasets It goes on to study the ability of the ICB to measure a network’s ability to fit random labels, and to distinguish between natural and randomly re-labelled datasets, finding that the bound can distinguish between random and natural labels in some settings but not others. Finally, the work evaluates the correlation between the value of the bound and empirical generalization performance.

**Broader Impact Concerns:**

None.

**Requested Changes:**


Motivated by my comments on the weaknesses of the paper, there are three major categories of changes I would like to see in the paper before I would be comfortable recommending it for publication.

1. Please provide a more detailed description of how the latent variable Z is computed in infinite width networks and prove the validity of the mutual information estimator.
2. The discussion of empirical falsification needs to be rewritten to acknowledge whether this is aiming to verify the correctness of the proof (seeing as this result has not to my knowledge been formally presented in peer-reviewed proceedings) or the validity of the assumptions.
3. I would also like to see a more detailed empirical analysis that explains how the observed correlation between the bound and generalization performance arises. For example, how  does generalization under interventions on different meta-parameters correlate with the generalization bound along the lines of [Dziugaite et al., 2020] — is ICB better at predicting generalization under some hyper-parameter selection problems than others?

**Strengths And Weaknesses:**

## Strengths
- The bound considered by the paper is not one that has been studied in great depth in the past -- most work on empirical generalization bounds has focused on those based on flatness of the local minimum, rather than looking at compression of the input space.
- The paper considers a number of different datasets which are reasonably diverse (though all vision), and identifies interesting differences in the correctness and predictive power of the proposed bound between these datasets.
- The work further studies a number of interesting experimental settings and asks thoughtful questions about the proposed bound, in particular: 1) evaluating effect of random labels on the bound, and 2) studying the predictive power, as opposed to correctness, of the bound on real architectures.
- The presentation and writing in the paper is polished — typos are not an issue, and the text flows nicely.


## Weaknesses

- There is significant lack of clarity in how the practical instantiations of the bound are computed: in particular, the work claims to evaluate the MI between a feature and the input, but also claims to use the NTK regime, in which to the best of my understanding networks don’t do feature learning.
- There are further potential issue with the MI estimation procedure, as the features on the training set are themselves a function of the training set, so estimating $I(X, Z_\theta)$ shouldn’t use the training set $X$ because of the dependence of the parameters $\theta$ (and hence the features $Z_\theta$) on $X$. It is possible that the procedure used gets around this issue but I wasn’t able to determine this from the information provided in the paper.
- The assumptions required for the ICB bound are not explicitly stated, and based on the cited paper they are in fact crucial for the validity of the bound. The paper should include a discussion of these assumptions and also evaluate the degree to which they are violated in the different datasets. This seems particularly relevant as the justification for ICB seems to be a limiting argument, and it is not clear how this scales with the dimension and number of samples in the problem. It is further unclear whether the generalization bound as presented in the paper is valid for any finite problem.
- The previous point is crucial and relates to a more philsophical issue I had with the paper: a generalization bound is either correct or not, and therefore there is no reason for it to be empirically “falsified”. The correctness of the bound stems from the correctness of its proof, and empirical investigation should not be necessary. An empirical falsification of a bound thus indicates one of two things: either the proof of the bound is incorrect, or the assumptions of the bound do not hold in the setting studied.
The study of infinite-width networks is not well-motivated: why not also include deep neural networks of finite width, as it is in these settings where feature learning, and hence input compression, will play a more significant role in the learning process?
If the bound is not in fact a provable upper bound on the generalization performance of a model but rather a heuristic, then a meaningful contribution to the literature would require a more detailed empirical analysis of the heuristic as the current empirical investigation is limited in the scope of problems considered and the detail of the analysis. This work only appears to consider binary classification problems, for example, and the differential performance in predicting generalization error and robust generalization error isn’t explained or motivated: what does robust generalization tell us about the ICB that classical generalization doesn’t?

---

> ### Author Response · Authors · 2022-10-12
> **Initial response to g7Gf (1 of 3)**
>
> We thank the reviewer for identifying several merits of our submission. We address only the list of weaknesses below. A new PDF will be uploaded shortly pending the outcome of the discussion and once we've finished responding to the more recently posted reviews.
>
> - There is significant lack of clarity in how the practical instantiations of the bound are computed: in particular, the work claims to evaluate the MI between a feature and the input, but also claims to use the NTK regime, in which to the best of my understanding networks don’t do feature learning.
>
> It is correct that there is no feature learning in this setting as the NTK depends only on the neural architecture. We agree that the word “feature” invites ambiguity as to whether features are learned from data or not, however, please note that we used the term “representation” in our submission. For the sake of generality, we let $Z$ refer to the output of any neural network layer in the Background section. To evaluate the input compression bound (ICB) in practice, however, we take $Z$ as the final output defined by the NTK, consistent with Shwartz-Ziv & Alemi (2020). We have clarified this in text in Section 2.1.
>
> - There are further potential issue with the MI estimation procedure, as the features on the training set are themselves a function of the training set, so estimating I(X; Z_\theta) shouldn’t use the training set X because of the dependence of the parameters \theta (and hence the features Z_\theta) on X. It is possible that the procedure used gets around this issue but I wasn’t able to determine this from the information provided in the paper.
>
> As clarified above, we use the NTK, therefore, “features” depend only on the neural architecture rather than being learned from data. Since we evaluate a generalization bound it is essential that all quantities in the bound be evaluated on training data only.
>
> - The assumptions required for the ICB bound are not explicitly stated, and based on the cited paper they are in fact crucial for the validity of the bound. The paper should include a discussion of these assumptions and also evaluate the degree to which they are violated in the different datasets. This seems particularly relevant as the justification for ICB seems to be a limiting argument, and it is not clear how this scales with the dimension and number of samples in the problem. It is further unclear whether the generalization bound as presented in the paper is valid for any finite problem.
>
> We agree that these assumptions are important and were absent from our initial submission. We have stated the assumptions in the revised draft. As implied by the reviewer, ICB relies on a limit (“for large enough input dimension ‘d’”), but it is hard to tell how close we are to this limit or if the conclusion holds in practice. Note: ICB also relies on some impossible-to-verify assumptions (see our Meta response).
> Therefore, empirical testing is needed to see if the theoretical predictions hold in practice for finite sizes. It is also possible that some of these assumptions are only required to derive the bound using known mathematical tools and that its conclusions may still hold without them.
>
> - The previous point is crucial and relates to a more philsophical issue I had with the paper: a generalization bound is either correct or not, and therefore there is no reason for it to be empirically “falsified”. The correctness of the bound stems from the correctness of its proof, and empirical investigation should not be necessary. An empirical falsification of a bound thus indicates one of two things: either the proof of the bound is incorrect, or the assumptions of the bound do not hold in the setting studied.
>
> We no longer use the term “empirical falsification”. As identified by the reviewer, the assumptions behind the bound are indeed critical, but challenging and even impossible to verify for finite samples from an unknown distribution. Given that it is difficult to verify the assumptions, as well as the bound’s probabilistic nature, we believe empirical work is essential to assess whether ICB is practically useful. We have revised the text to clarify that we check for instances where the bound may be violated and whether it is useful in practical settings.

---

> ### Author Response · Authors · 2022-10-12
> **Initial response to g7Gf (2 of 3)**
>
> - The study of infinite-width networks is not well-motivated: why not also include deep neural networks of finite width, as it is in these settings where feature learning, and hence input compression, will play a more significant role in the learning process?
>
> We thank the reviewer for the suggestion. It would be great to study finite-width DNNs as well as their infinite-width counterparts. Unfortunately, the mutual information $I(X; Z)$ is ill-defined in standard DNNs, and is notoriously challenging to estimate accurately even when networks are adapted to make MI well-defined (Goldfeld et al., ICML 2019). For instance, this may require specialized architectures (e.g., see Gabrie et al., NeurIPS 2018) with different metaparameter and dataset limitations than are associated with our methodology. For these and other resource limitations a combined 12-page presentation on infinite- and finite-width networks is not practical.
>
> Input compression is present in linear models, and is therefore not limited to deep learning. For instance, in the Gaussian information bottleneck (GIB) task (Chechik et al., JMLR 2005), the optimal solution was shown to be a noisy linear projection to eigenvectors of the normalized regression matrix, and a compression parameter, $\beta$, determines the dimension and scale of each eigenvector. As compression behaviour is already non-trivial for linear regression, we believe our work represents an important and exciting extension to kernel regression.
>
> As mentioned in the Introduction and suggested by the title, we intentionally limited the scope of this work to infinite-width networks to ensure the accuracy of MI estimates. This enabled us to study the ICB in a practical setting for the first time to the best of our knowledge. We believe this is a crucial first step in the absence of empirical work on the ICB to date. Our scope is further motivated by the fact that  several generalization phenomena in kernel regression and deep learning are shared, yet not quite captured by existing theories (Belkin et al., 2018; 2019).
>
> We discussed the extension of this work to deep learning and associated issues in more detail in the section “Relevance to deep learning” of the “Discussion” and in “Input compression and deep learning” of the “Related Work”. We have revised the Introduction to clarify that MI is ill-defined in standard DNNs to better motivate our approach. However, we believe that we have been forthright and justified in limiting our scope to infinite-width networks.
>
> - If the bound is not in fact a provable upper bound on the generalization performance of a model but rather a heuristic, then a meaningful contribution to the literature would require a more detailed empirical analysis of the heuristic as the current empirical investigation is limited in the scope of problems considered and the detail of the analysis.
>
> We believe that the ICB is a correct PAC generalization bound. We also believe that our empirical analysis is substantial enough to demonstrate its practical utility: We evaluated 9,000 models for each of five image datasets (45,000 total) by drawing different random seeds, metaparameters, and task labels. Many works on generalization merely aim to show that a bound is non-vacuous or near the empirical GE in specific settings. We have exceeded this standard by exploring metaparameter configurations and searching for potential bound violations from the time of initialization to steady-state on five datasets, training on random labels, as well as investigating the bound’s sensitivity to corrupted inputs. Of course our investigation is not exhaustive with respect to all questions one may ask, but we hope and expect that our study encourages others to further examine the advantages and limitations of ICB, which we believe has been underexplored.

---

> ### Author Response · Authors · 2022-10-12
> **Initial response to g7Gf (3 of 3)**
>
> - This work only appears to consider binary classification problems,...
>
> Please note that there is considerable precedent for studying binary classification in the generalization literature (e.g., Dziugaite and Roy, UAI 2017; Shwartz-Ziv & Alemi PMLR 2020).
>
> We provided a reason for this restriction in section “3 Experiments”:
>
> >“We focus on binary classification like much of the generalization literature, which also enables us to more efficiently evaluate MI bounds by processing kernel matrices that scale by $N_\text{trn}^2$ rather than $(k \times N_\text{trn})^2$ for $k$-classes. Aside from this computational advantage, our methodology extends to the multi-class setting.”
>
> We would not have been able to investigate the three questions in as much depth had we focused on multi-class classification. Furthermore, our approach increases the number of distinct classification tasks investigated by a factor of 9 (Exp. A) and 45 (Exp. B), leading to a finer-grained analysis of where the bound holds. For example, ICB was satisfied by $100%$ of models ($N=2534$) on six of nine EuroSAT tasks, with two tasks in particular reducing the overall average. This inter-task discrepancy would have been obscured under the original multi-class problem. Our analysis opens the door to future investigation of inter-task differences for these datasets. Since these were the same tasks for which ICB had difficulty detecting random training labels, it may be that they present a lower signal-to-noise ratio than other tasks, but this is merely a guess.
>
> - the differential performance in predicting generalization error and robust generalization error isn’t explained or motivated: what does robust generalization tell us about the ICB that classical generalization doesn’t?
>
> In some cases, ICB may not be sensitive enough to detect small differences in the standard GE where it otherwise detects changes in robust GE. Changes in robust GE imply a difference in the mean prediction margin, or distance to the decision boundary. As shown by Figure 1, adding a small amount of noise to the inputs reveals a closer relationship between ICB and robust GE than for standard GE. The purpose of these experiments is to understand whether ICB better explains standard GE or robust GE, and that minimizing ICB may be beneficial even when standard GE is already small.
>
> __Requested Changes__
>
> 1. Please provide a more detailed description of how the latent variable Z is computed in infinite width networks and prove the validity of the mutual information estimator.
>
> As mentioned above, $Z$ is the predictive output on the training set. Mutual information bounds used in this work were first derived using an approximation by van den Oord et al., (2018) and subsequently re-derived without approximation by Poole et al., (ICML 2019), therefore we did not see a need to duplicate this work. The methodology of incorporating the NTK predictive distribution into these MI bounds was also used previously by Shwartz-Ziv & Alemi (PMLR 2020). We have made the discussion of each of these points explicit in the revision.
>
> 2. The discussion of empirical falsification needs to be rewritten..
>
> We no longer use this term. Please refer to our “Meta response Re: falsification and assumptions of the input compression bound (reviewers g7Gf and GTi6)”
>
> 3. I would also like to see a more detailed empirical analysis that explains how the observed correlation between the bound and generalization performance arises... along the lines of [Dziugaite et al., 2020] — is ICB better at predicting generalization under some hyper-parameter selection problems than others?
>
> We thank the reviewer for this suggestion. We have performed *coupled-network* experiments as per Dziugaite et al., (NeurIPS 2020) to assess ICB’s ability to predict generalization under specific metaparameter interventions. We included three metaparameters that were excluded by Dziugaite et al.: NTK diagonal regularization, training time, and activation function. We omitted width, learning rate, and mini-batch size, as these do not apply to our setting.
>
> Intervening on the number of training samples, diagonal regularization, and training time yielded consistently small sign-error. Therefore, these three metaparameters strongly influence the overall correlation between ICB and GE. Conversely, the activation function and depth had large sign errors. More details about this experiment including numerical results, sample sizes, and metaparameter values may be found in a new Appendix section titled "Metaparameter study".

---

### Review · Reviewer_GTi6 · 2022-10-03

**Summary Of Contributions:**

In this paper, the authors empirically explore an MI-based bound to upper bound generalization error. Their theoretical generalization error bound is based on a compression technique that relates input and final representations. Experimental evidence across a few architectures and datasets shows the applicability of this bound in practice.

**Requested Changes:**

Please see the comments in the weakness section above. Addressing all of those bullets would be crucial to securing an acceptance. Moreover, to make the work complete it would be great to include the following (not necessary to secure acceptance):

- Formal proof for equation 4 should be included in the paper. While authors cite Shwartz-Ziv 2019, a formal proof for equation (4) with consistent terminology can help readers as the paper primarily uses that bound.



**Strengths And Weaknesses:**

**Strength**:
- This paper presents empirical analysis of an existing bound. This is particularly important as it can highlight how bounds derived under restrictive settings behave more generally.

**Weakness**:
- The abstract highlights that "in an attempt to falsify the bound", it would be great to explictly list down all the reasons on why one should expect the bound to incorrect. This should include the setting under which the bound is theoretically applicable and clearly highlight how they may not be true in practice.

- The language in abstract about "In our attempt to empirically falsify the theoretical bound," is a bit too strange. To the best of my understanding this paper attempts to emperically study a bound derived theoretically under some strong assumptions. If this understanding is correct, calling it as trying to empirically falsify is incorrect language in my opinion as the goal is not to falsify the bound in the settings it was derived in.

- Figure 1 shows the bound with shaded grey region. It is unclear why the bound is shown with a shaded region. Shouldn't it be simply a line? While it may mean that the test error can lie anywhere in between, the current exposition is not highlighing this clearly.

- Figure 2, and more broadly, experiments in that figures seems a bit weird. While to theoretically verify the bounds, it makes sense to have values below the y =x line, it is unclear why it is the only criterion used in the paper. A loose bound, as loose as predicting an error of 100% would always be correct according to the `satisfaction' criterion used in Figure 2. To understand the tightness of the bound, should the bound be close to the line y =x.

- Since one of the goal of the paper is find bounds that correlate well with GE, paper should include discussion (and perhaps compare) with other related work that leverages unlabeled data to obtain bounds [1,2]

- Moreover, a discussion with other mutual information bounds must be inluded in the paper [3,4,5].

- Contribution of this work also a bit skim. First, it is unclear why authors only focus on the bound in equation (4), what are the (desirable) properties of this bounds that even motivates studying only this bound. Second, it would be great if authors can add comparison to other MI based bounds derived in other related work highlighted above.


References:
[1] Garg et al. RATT: Leveraging Unlabeled Data to Guarantee Generalization. ICML 2021.
[2] Jiang et al. Assessing Generalization of SGD via Disagreement. ICLR 2022.
[3] Achille and Soatto. Emergence of Invariance and Disentanglement in Deep Representations. JMLR 2018.
[4] Harutyunyan et al. Improving Generalization by Controlling Label-Noise Information in Neural Network Weights. ICML 2020.
[5] Xu and Raginsky. Information-theoretic analysis of generalization capability of learning algorithms. NeuRIPS 2017

---

> ### Author Response · Authors · 2022-10-03
> **Quick clarification re: Reference [5]**
>
> Hi, just a quick note that reference [5] is absent from the list of references, do you mind clarifying the citation here? Thank you.

---

> > ### Comment · Reviewer_GTi6 · 2022-10-04
> > **Fixed**
> >
> > Thanks for the quick catch, fixed.

---

> ### Author Response · Authors · 2022-10-14
> **Initial response to GTi6 (1 of 2)**
>
> We thank the reviewer for their feedback. We address only the list of weaknesses below. A new PDF will be uploaded shortly pending the outcome of the discussion and once we've address all reviews.
>
> - The abstract highlights that "in an attempt to falsify the bound" …The language in abstract about "In our attempt to empirically falsify the theoretical bound," is a bit too strange.
>
> We agree with you and reviewer g7Gf that the formulation “empirical falsification” was awkward and have changed it accordingly. Our aim is to assess how well ICB works in practice. This has been clarified in text. The language arose in the sense that, although we believe in the practical utility of this bound, rather than solely trying to confirm our intuition with experiments that support its effectiveness, we aim to uncover bound violations, e.g., where its assumptions may not be satisfied in important ML settings.
>
> Please refer to our *Meta response* for more detail.
>
>
> - Figure 1 shows the bound with shaded grey region. It is unclear why the bound is shown with a shaded region...
>
> We use a shaded region to indicate that generalization errors anywhere within this region are consistent with the bound. We have clarified this in text. We believe that this is a reasonable stylistic choice and that adding a fourth line to this plot may be confusing.
>
> - Figure 2, and more broadly, experiments in that figures seems a bit weird. While to theoretically verify the bounds, it makes sense to have values below the $y=x$ line, it is unclear why it is the only criterion used in the paper.
>
> Checking that the empirical GE is less than the theoretical bound is merely the first of several criteria that we use to assess the ICB, as outlined in itemized list 1-3 of the Introduction. Tightness of the bound and its correlation with GE are assessed in Sections 4.3 “Vacuous or non-vacuous?” and 4.4 “Relationship between theoretical bound and generalization error”.
>
> Re “To understand the tightness of the bound, should the bound be close to the line y =x”, we believe that this is a fair criterion for a generalization error (GE) *estimator*, but not necessarily for a GE *upper bound*. A model may generalize well by chance even though the bound may be large. As illustrated by Figure 1, a loose bound may also reflect that examples are classified with a small prediction margin and that a model has poor robustness. Nonetheless, we agree that a bound should be tight under at least some metaparameter settings; this is the topic of Section 4.3 “Vacuous or non-vacuous?".
>
> - First, it is unclear why authors only focus on the bound in equation (4), what are the (desirable) properties of this bounds that even motivates studying only this bound. Second, it would be great if authors can add comparison to other MI based bounds derived in other related work highlighted above.
>
> We study the ICB because: 1) To the best of our knowledge it has never been studied before empirically; 2) All other bounds we are aware of are *model-centric* whereas this bound is *data-centric*. In this vein, reviewer g7Gf has also noted that:
> > The bound considered by the paper is not one that has been studied in great depth in the past -- most work on empirical generalization bounds has focused on those based on flatness of the local minimum, rather than looking at compression of the input space.
>
> Given that all other model-centric bounds fail to be robust under at least one metaparameter intervention (Dziugaite et al.,NeurIPS 2020), we thought it promising to study an overlooked bound constructed by a fundamentally different approach. We provided this line of reasoning in the introduction. The use of the Information Bottleneck as an underlying principle relevant to generalization in deep learning has also been subject to considerable debate (Saxe et al., ICLR 2018; Goldfeld et al., ICML 2019). We believe that our work adds valuable evidence to this body of work by strengthening the connection between input compression and generalization in a deep learning-adjacent model family of infinite-width networks.
>
> - [The] paper should include discussion (and perhaps compare) with other related work that leverages unlabeled data to obtain bounds [1,2]. Moreover, a discussion with other mutual information bounds must be inluded in the paper [3,4,5].
>
> We thank the reviewer for the suggestion, we are still reading papers [1-5] and will provide an update shortly.

---

> ### Author Response · Authors · 2022-10-20
> **Initial response to GTi6 (2 of 2)**
>
> - Since one of the goal of the paper is find bounds that correlate well with GE, paper should include discussion (and perhaps compare) with other related work that leverages unlabeled data to obtain bounds [1,2]
>
> We thank the reviewer for pointing us to this important work. We have discussed them under the heading __Generalization bounds from unlabeled data__ of the Related work.
>
> - Moreover, a discussion with other mutual information bounds must be included in the paper [3,4,5].
>
> We agree that these works are relevant. We have discussed them under the heading __Information compression and generalization__ of the Related work.
>
> Like ICB, they connect the mutual information between *inputs* and *outputs* to generalization error. Whereas ICB is evaluated w.r.t. to a single *model*, these works consider *learning algorithms*. Aside from their important theoretical contributions, Achille and Soatto [3] and Harutyunyan et al. [4] focus their empirical investigations on *optimizing* mutual information (e.g., via the Information Bottleneck lagrange multiplier $\beta$) to prevent memorization of random labels and/or to improve test accuracy. This differs from our objective to empirically test a GE bound (ICB). Achille and Soatto [3] connect the MI $I(w; D)$ to a PAC bound which they admit is known to be vacuous and do not evaluate empirically.
>
> Our work builds upon that of Shwartz-Ziv and Alemi (PMLR 2020), who evaluated [in their Appendix C.7] the MI between parameters ($\theta$ rather than $w$ using their notation) and a dataset $D$ in relation to generalization for infinite-width networks. Importantly, they remark that:
> > The MI between the parameters and the dataset $I(\theta; D)$ has been shown to control for overfitting (Bassily et al., 2017) [...] and to provide generalization bounds in PAC-Bayes frameworks (Achille and Soatto, 2017).
>
> > $I(\theta; D)$ tends to infinity as the time goes to infinity. This renders the usual PAC-Bayes style generalization bounds trivially vacuous for the generalization of infinitely wide neural networks at late times. Yet, infinite networks can generalize well (Arora et al., 2019).
>
> Hence, we focused on a previously untested generalization bound relying on the mutual information between the input $X$ and its final representation $Z$, $I(X; Z)$, which does not increase indefinitely as time goes to infinity.

---

### Review · Reviewer_G5hL · 2022-10-06

**Summary Of Contributions:**

This paper empirically evaluates Input Compression Bound (ICB) as a generalization bound. The ICB directly links the generalization with the mutual information between input X and representation Z. Through three axes: tightness of the bound (vacuous or non-vacuous), percentage of trials satisfying the bound, and ranking generalization errors, the authors show that: for many datasets, the ICB has a good satisfaction rate (although for some have low satisfaction rate); randomization test may be used to help identify when ICB performs bad as a GE bound for a variety of models; it is non-vacuous; etc. Overall, the extensive empirical study, the authors show that ICB is generally effective and useful for indicating generalization error.

**Broader Impact Concerns:**

There is no need for Broader Impact Statement.

**Requested Changes:**

Corresponding to the weaknesses above, (2) is critical for the acceptance; and deeper explanation for (3), and (1) comparing with other generalization bounds can significantly strengthen the work.

Minor:
1. Can the authors explain in the text or Appendix how the \lambda regularization is used? It is not clear in current manuscript.

**Strengths And Weaknesses:**

Strengths:

(1) The problem this paper addresses is important, and the experiment result is interesting, which can provide good materials for further discussions in the community.

(2) The experiments are performed in a quite thorough way and to my knowledge, is sound.

Weaknesses:

(1) It may be useful to compare with some other generalization bounds as mentioned in the related works. Even though it is difficult as said by the authors, it may still be useful to see the relative strengths of different generalization bounds.

(2) The writing of the experiment section can be more clear. Right now it seems there are many observations, but not presented in a clear and well-organized way.

(3) For some datasets (e.g. SVHN, CIFAR), the ability of ICB to bound generalization error (measured in satisfaction rate) is low. The author observe that these are also the ones that can be identified via randomization test. Does the authors have an intuitive explanation/hypothesis for this? If so, the authors can also perform experiments to verify or disprove the hypotheses. In this way, we can understand deeper about the failure modes of ICB, and may even improve it.

---

> ### Author Response · Authors · 2022-10-30
> **Initial response to G5hL**
>
> We thank the reviewer for their feedback and noting the importance of our work and strength of experiments. We address weaknesses below.
>
> - (1) It may be useful to compare with some other generalization bounds...
>
> We made a strategic decision to conduct several different experiments on a previously untested generalization bound, and appreciate that comparisons to other generalization bounds are helpful. Many existing theories of generalization have been considered by others, e.g., in the framework of Dziugaite et al., (2020). Our revised Section 4.4 now adopts this framework, therefore readers may contrast ICB with any of the metrics considered by Dziugaite et al., (2020), which may depend on the metaparameters to be intervened upon as well as the application. Dziugaite et al. also mention that measures performing well on their benchmark “would not necessarily serve to be useful in generalization bounds”. Assessing the utility of ICB as a *generalization bound* is our main objective, for which Section 4.4 serves as one of several criteria.
>
> We discuss a related information-theoretic PAC-Bayes bound in our comment to reviewer GTi6 (https://openreview.net/forum?id=jbZEUtULft&noteId=y9NZPAkMJP) along with rationale as to why we do not evaluate it directly given that it is vacuous. We compare numerical results to a variant of this bound by Dziugaite and Roy (UAI 2017) in Section 4 as well as in the Discussion.
>
>  - (2) The writing of the experiment section can be more clear...
>
> We thank the reviewer for this feedback. We have simplified the experiments by replacing procedures Exp. A and B with a single procedure in the revised draft, and condensed the results to more clearly emphasize key observations.
>
> - (3) For some datasets (e.g. SVHN, CIFAR), the ability of ICB to bound generalization error (measured in satisfaction rate) is low. The author observe that these are also the ones that can be identified via randomization test. Does the authors have an intuitive explanation/hypothesis for this? If so, the authors can also perform experiments to verify or disprove the hypotheses. In this way, we can understand deeper about the failure modes of ICB, and may even improve it.
>
> We now have a better understanding of the ICB satisfaction rates for SVHN and CIFAR, which have improved considerably in the latest draft. Quoting from the new results section 4.2:
>
> > __Architecture__ type has a significant effect on ICB satisfaction rate: CNNs had an $11$-$14$% greater ICB satisfaction rate than MLPs for CIFAR, and $10$-$18$% greater for SVHN, where the range depended on the loss function. Intriguingly, the superior ICB satisfaction rate of CNNs was not due to greater test-set accuracy. CNNs had lower GE as a result of their lower *train*-set accuracy (Table A5).
>
> As a result, the ICB satisfaction rate is now at 95% overall for SVHN for the MSE loss function, and the number of training samples played a major role for the 0-1 loss, which satisfied ICB at less than 95%.
>
> We agree it would be great to have a better understanding of the randomization test in relation to the ICB satisfaction rate. We don’t wish to advance a new hypothesis at this time, which we lack space/time to thoroughly evaluate. Nonetheless, we stand by the effectiveness of this test and believe it to be actionable for practitioners given that it only requires the use of training data. Under the new simplified procedure based on CNNs for CIFAR-10, the randomization test also increased in effectiveness with $\tau = 0.87$ versus $0.65$ in the previous version (Table 3).
>
> - Can the authors explain in the text or Appendix how the \lambda regularization is used? It is not clear in current manuscript.
>
> We thank the reviewer for the suggestion and have added a brief explanation about diagonal regularization in section 3 Experiments.

---

### Review · Reviewer_egUC · 2022-10-11

**Summary Of Contributions:**

This work studies the problem of generalization error estimation without the use of validation samples. Authors studied the mutual information (MI) between inputs and final layer representations of deep neural networks, and derive a so-called input-compression bound of generalization errors ($GE_\mathrm{ICB}$). Then, authors carried out empirical studies to investigate the correlation between MI and the generalization error.  Specifically, authors conducted several sets of experiments to investigate the nexus between theoretical GE bounds and the generalization errors. The results demonstrate the feasibility of using ICB to estimate generalization errors.

**Requested Changes:**

Please consider to include the three points in the weakness section for revision.

**Strengths And Weaknesses:**

Strengths
1. Authors studied an interesting yet importance problem in the field.
2. Authors clearly stated the derivations of $GE_{ICB}$ from the statistical learning theory (e.g., PAC) and information theory.
3. Several experiments have been done to demonstrate the feasibility of the proposal.

Weakness.
1. Eq. (3) should hold in a probability higher than $1-O(\delta)$ or $1-\delta$. So, how the confidence factor $\delta$ would influence the estimation? If ``ICB satisfaction rate'' equals to $1-\delta$, please make it clear in the texts. I may suggest you to complete the inequality in Eq. (3) with the probability term and introduce the detailed definition of $\delta$ there. Another issue is that the incorporation of $\delta$ makes ICB a tradeoff between mutual information (input-output) and the confidence levels of GE estimation. Please discuss this issue in the revision.

2. Experiments results are intensive and verbose. Could you please enumerate the hypotheses or expected observations you hope to see throughout your experiments, so as to test your major hypothesis on ICB and GE? Please highlight your claims from your experiments.

3. Authors discussed the correlations between ICB and GE, and mentioned the NeurIPS competition on generalization error predictions. I am wondering if it is possible to include more competitors from the competition to compare with ICB? I understand ICB is a more comprehensive tool with confidence bounds.

---

> ### Author Response · Authors · 2022-10-31
> **Initial response to egUC**
>
> We thank the reviewer for their comments. We respond to weaknesses below.
>
> - 1. Eq. (3) should hold in a probability higher $1 - O(\delta)$ or $1 - \delta$. So, how the confidence factor would influence the estimation? (...)
>
> The $\delta$ parameter is inherited from the PAC framework and accounts for a small probability that the training set received by a learning algorithm is misleading (i.e., not representative of the data-generating distribution) or the finite samples do not reflect all relevant details of the distribution (Shalev-Shwartz and Ben-David, (2014)).
>
> We have stated at the beginning of Section 4.1:
>
> > We refer to the percentage of tuples (ICB, GE) for which GE < ICB as the “ICB satisfaction rate''... We expect $\approx 95\\%$ of samples to satisfy this property as ICB is evaluated at $95\\%$ confidence ($\delta=0.05$).
>
> When we decreased $\delta$ to $0.01$ from $0.05$ to evaluate ICB at $99\\%$ confidence, this increased the ICB satisfaction rate by $6.2\\%$ for SVHN on the 0-1 loss, compared to an expected $\approx 4%$ difference ($99\\%-95\\%$). We also tried *increasing* $\delta$ to $0.2$ for $80\\%$ confidence, thus making the ICB smaller which is expected to *decrease* the ICB satisfaction rate. Indeed, this change decreased the ICB satisfaction rate by $11.2\\%$, compared to an anticipated difference of $\approx 19\\%$ ($95\\% - 80\\%$). We are thrilled that ICB is close enough to the empirical GE to be affected by $\delta$; this confidence parameter is often ignored in empirical studies or left at a default $95\\%$ setting.
>
> Results were less sensitive to $\delta$ for CIFAR-10 and EuroSAT ($\approx 1\\%$ difference for $99\\%-95\\%$), and independent from $\delta$ for FashionMNIST as the compression term alone (in the Eq. (4) numerator) sufficed to bound the small GEs for this dataset.
>
> Please let us know if this answers your question? We can include these results in the paper if the reviewer believes they would be helpful, as we are mindful that the paper already contains many experiments (see next item 2).
>
> - 2. Experiments results are intensive and verbose (...)
>
> We thank the reviewer for this feedback. We have simplified the experiments by replacing methods Exp. A and B with one procedure in the revised draft, and condensed the results to more clearly emphasize key observations. Fortunately, the improved results using CNNs and greater train-set sizes also simplify the subsequent analyses in support of ICB as an effective tool.
>
> - 3. Authors discussed the correlations between ICB and GE (...) I am wondering if it is possible to include more competitors from the competition to compare with ICB? I understand ICB is a more comprehensive tool with confidence bounds.
>
> Please see the following comments to [G5hL](https://openreview.net/forum?id=jbZEUtULft&noteId=avp6uW03TB) and [GTi6](https://openreview.net/forum?id=jbZEUtULft&noteId=y9NZPAkMJP) for rationale as to why we did not numerically evaluate these other bounds again ourselves which were considered by the NeurIPS competition and Dziugaite et al. (2020).

---

> > ### Comment · Reviewer_egUC · 2022-11-11
> > **Responses and revisions have addressed my concerns.**
> >
> > It is very nice to see information-theoretic estimators (input-output compression) could interplay with PAC learning to formulate a practical generalization bound with confidence intervals for deep neural networks. The background section introduced the intution fo ICB and how it connects with previous works. Experiments and additional results in appendix demostrated the feasibility of using ICB as an estimator for generalization error bounds.
> >
> > From an empirical studies' perspective, authors have already provided evidences to show that (1) the hypothesis of "ICB as a (practical, maybe) generalization error bound estimator" is well formulated and intuitive, as it is derived from existing works/investigation; (2) ICB works in a number of experiments; (3) authors also conduct a number of experiments with factors controlled (such as the choice of \delta, DNN architectures and datasets) in various settings to validate their findings.

---

### Author Response · Authors · 2022-10-12
**Meta response Re: falsification and assumptions of the input compression bound (reviewers g7Gf and GTi6)**

__Re: Reviewer g7Gf’s requested change #2:__
> “The discussion of empirical falsification needs to be rewritten to acknowledge whether this is aiming to verify the correctness of the proof (seeing as this result has not to my knowledge been formally presented in peer-reviewed proceedings) or the validity of the assumptions”

__Re: Reviewer GTi6’s comment:__
> “The abstract highlights that ‘in an attempt to falsify the bound’...”

We agree that the terminology "empirical falsification" was not clear enough and led to some confusion which we would like to carefully address. Our ultimate goal is to test how useful the Input Compression Bound is in practice, therefore we emphasize that this is an empirical work. We take the ICB as given (by the referenced work by Tishby (2017); Shwartz-Ziv et al. (2019)), and while the derivation of the ICB and the proof of the associated theorem appear reasonable to us, a formal review of the proof is not our goal.

It is impossible to check the assumptions directly because they assume something about the data-generation process, which we can not access from finitely many samples in the given dataset. We believe that our work actually falls into a third category, not quite captured by the binary choice between verifying the proof or verifying the assumptions presented by reviewer g7Gf: We treat the ICB as a tool, and we empirically test how useful this tool is in practice.

It is also possible for the bounds’s conclusions to hold without all of the technical assumptions used in the proof. For instance, one assumption is that the probability associated with the input variable $X$ is assumed to obey the ergodic Markov random field structure. A Markov Random Field is an undirected graphical model, used to model data distributions with particular conditional independency structure, which is commonly used for spatial data, including images (see Chapter 19 in Kevin Murphy’s *Machine Learning: a Probabilistic Perspective* book for an excellent introduction). Mathematically, this means that $p(x)$ factorizes into a product of terms which represent the potentials for each clique on the underlying graph. In terms of correlations, this means that each pixel is strongly correlated with its immediate neighbors, but not with pixels that are further away.

The “ergodic” part is essential for the derivation of the ICB: An ergodic Markov Random Field (MRF) does not “get stuck” in any part of the state space; in other words, there is a nonzero probability for every possible state to be reached. Ultimately, this is the necessary assumption to invoke the Asymptotic Equipartition Property (AEP), which in turn allows us to argue about typicality. Defining the typical set is the crux of the ICB derivation, because it enables us to quantify the hypothesis space cardinality in terms of entropy. From here, the rest follows from information-theory fundamentals.

As our experiments do not involve testing the ergodic MRF assumption or the typicality, we argue that our work is not about validating assumptions, but rather assessing ICB as a practical tool.

---

### Author Response · Authors · 2022-10-21
**Updated PDF and new result re ICB satisfaction rate of CNNs**

We have updated the PDF in accordance with our initial responses to reviewers GTi6 and g7Gf, including some minor issues identified by the other reviewers.

Aside from this, following g7Gf's comments and our investigation into Dziugaite et al., (NeurIPS 2020), we have discovered that the training set size and architecture have a considerable impact on the ICB satisfaction rate (ICB Sat., discussed in S4.1). In particular, CNNs have a considerably higher ICB satisfaction rate than fully-connected models for SVHN and CIFAR-10. Assessing generalization error (GE) in terms of MSE, ICB Sat. improved from 72.6% (our current Fig. 6) to __95.2%__ overall for SVHN, and from 48.7% (our current Fig. 6) to __80.5%__ for CIFAR-10. ICB Sat. increased similarly when GE was expressed in terms of 0-1 error, e.g., 59.3% (our current Fig. 2) to 84.9%.

As a result, we are currently finalizing changes to Fig. 2 and potential downstream impacts.

---

### Author Response · Authors · 2022-11-03
**Updated PDF and happy to provide any clarifications**

Dear reviewers, thank you for your comments. We have completed the revision including new CNN results on Oct. 30th and a more streamlined experimental procedure. As the PDF is stable again and we believe we have addressed all reviewer comments, we would appreciate the opportunity to provide any clarifications before the end of the discussion phase. Thank you, -paper 282 authors

---

### Decision · Action_Editors · 2022-12-03

**Recommendation:** Accept as is

**Comment:**

The authors have engaged with the reviewers and through rebuttals, addressed reviewers' concern in a satisfactory way. On a personal note, I found the manuscript to be an enjoyable read, and I appreciate the care the authors have put into explaining their results. I expect the camera-ready to reflect the discussions with reviewers. Congratulations on a fine piece of work and for your contribution to TMLR!

**Audience:**

I believe an empirical study of the generalisation error of (infinite-width) neural networks will be of great interest to the TMLR community -- and beyond.

**Claims And Evidence:**

All claims are supported by convincing and clear evidence.